# Coordinated head direction representations in mouse anterodorsal thalamic nucleus and retrosplenial cortex

**Marie-Sophie H van der Goes[1], Jakob Voigts[1,2,3], Jonathan P Newman[2,4], Enrique HS Toloza[1,5,6], Norma J Brown[1], Pranav Murugan[7], Mark T Harnett[1]\***

[1]Department of Brain & Cognitive Sciences, McGovern Institute for Brain Research, Massachusetts Institute of Technology, Cambridge, United States; [2]Open-Ephys Inc, Atlanta, United States; [3]HHMI Janelia Research Campus, Ashburn, United States; [4]Department of Brain & Cognitive Sciences, Picower Institute for Learning and Memory, Massachusetts Institute of Technology, Cambridge, United States; [5]Department of Physics, Massachusetts Institute of Technology, Cambridge, United States; [6]Harvard Medical School, Boston, United States; [7]Department of Electrical Engineering and Computer Science, Massachusetts Institute of Technology, Cambridge, United States

**Abstract** The sense of direction is critical for survival in changing environments and relies on flexibly integrating self-motion signals with external sensory cues. While the anatomical substrates involved in head direction (HD) coding are well known, the mechanisms by which visual information updates HD representations remain poorly understood. Retrosplenial cortex (RSC) plays a key role in forming coherent representations of space in mammals and it encodes a variety of navigational variables, including HD. Here, we use simultaneous two-area tetrode recording to show that RSC HD representation is nearly synchronous with that of the anterodorsal nucleus of thalamus (ADn), the obligatory thalamic relay of HD to cortex, during rotation of a prominent visual cue. Moreover, coordination of HD representations in the two regions is maintained during darkness. We further show that anatomical and functional connectivity are consistent with a strong feedforward drive of HD information from ADn to RSC, with anatomically restricted corticothalamic feedback. Together, our results indicate a concerted global HD reference update across cortex and thalamus.

**\*For correspondence:**
harnett@mit.edu

## Editor's evaluation

This useful study investigates the coordination of neurons coding for head direction in the anterior thalamus and the retrosplenial cortex during environmental manipulations. The evidence supporting the claims of the authors is solid. The paper will be of interest to neuroscientists working on spatial navigation.

## Introduction

In order to enable efficient navigation, internal representations of self-location and orientation must be updated as sensory experience and behavioral demands fluctuate. Changes in environmental information are known to trigger remapping of place (*Muller and Kubie, 1987*), grid (*Fyhn et al., 2007*) and realignment of HD receptive fields (*Taube, 1990*; *Goodridge et al., 1998*; *Knierim et al., 1998*). Visual cues influence the HD signal by providing an external anchoring reference (*Taube, 1990*; *Taube and Burton, 1995*), counteracting the drift from stochastic error in angular velocity integration observed

in darkness (*Mizumori and Williams, 1993*; *Stackman and Taube, 1997*; *Valerio and Taube, 2012*). In the insect and mammalian HD systems, realignment has been observed in the form of rotations of preferred firing directions (PFD) of HD cells in response to rotations of a prominent visual cue - often a color-contrast card (*Taube, 1990*; *Goodridge et al., 1998*), a narrow band (*Seelig and Jayaraman, 2015*) or scene (*Kim et al., 2019*) on an LED screen. Hebbian synaptic plasticity mechanisms, acting in specific circuit arrangements, have been proposed to explain these phenomena in the fly ellipsoid body (*Fisher et al., 2019*; *Kim et al., 2019*). Network models with similar architecture have been applied to the rodent HD system (*Hahnloser, 2003*; *Knight et al., 2014*; *Page et al., 2014*; *Skaggs et al., 1995*). However, unlike the fly brain (*Franconville et al., 2018*; *Hanesch et al., 1989*), the circuitry that drives HD realignment in the rodent brain is not yet resolved. Several studies have indicated that cortical regions play a role in the stability and the visual anchoring of HD cells (*Clark et al., 2010*; *Golob and Taube, 1999*; *Goodridge and Taube, 1997*), but whether the HD representation is first aligned to the sensory cues in cortex and then updated in downstream regions is still unknown. Understanding the dynamics and the connectivity of visual-HD integration is the first necessary first step to uncover the mechanisms that lead to realignment in the mammalian HD system.

Previous studies in the rat indicate that the HD code in the anterodorsal thalamic nucleus (ADn), the necessary thalamic relay of HD to the hippocampal formation and cortex (*Calton et al., 2003*; *Frost et al., 2021*; *Goodridge and Taube, 1997*; *Jenkins et al., 2004*; *Winter et al., 2015*), becomes unstable and is less likely to remap to reflect cue rotations after lesions to the retrosplenial cortex (RSC; *Clark et al., 2010*) or the post-subiculum (POS; *Goodridge and Taube, 1997*). Both of these regions are strongly interconnected with visual areas (*Sugar et al., 2011*; *Van Groen and Wyss, 2003*), with each other (*Kononenko and Witter, 2012*; *Wyss and Van Groen, 1992*), and with ADn (*Jankowski et al., 2013*). While HD dominates the spatial code in POS (*Taube, 1990*; *Peyrache et al., 2017*; *Laurens et al., 2019*), RSC exhibits diverse visuo-spatial activity (*Cho and Sharp, 2001*; *Fischer et al., 2020*; *Mao et al., 2017*; *Powell et al., 2020*; *Voigts and Harnett, 2020*), with complex receptive fields shaped by multiple spatial correlates (*Alexander et al., 2020*; *Alexander and Nitz, 2017*; *Jacob et al., 2017*). Unlike POS, which presents mostly homogeneous, cue-guided rotations of HD fields in response to rotations of a prominent visual cue (*Taube, 1990*), diverse responses of RSC HD cells to cue manipulations or specific environmental configurations have been reported (*Chen et al., 1994a*; *Jacob et al., 2017*; *Sit and Goard, 2023*), suggesting differential influence of idiothetic and allothetic cues and complex encoding schemes.

As a cortical association area, RSC plays a critical role in spatial cognition (*Knight and Hayman, 2014*; *Mitchell et al., 2018*; *Vann et al., 2009*) and spatial memory (*Miller et al., 2019*; *Miller et al., 2014*): rats and humans with RSC lesions show impairments in route planning as well as identification and flexible use of navigational landmarks (*Hindley et al., 2014*; *Maguire, 2001*; *Pothuizen et al., 2008*; *Vann and Aggleton, 2004*). Specifically, in the intact RSC, associations between egocentric and allocentric reference frames become evident with spatial tasks (*Alexander et al., 2020*; *Alexander and Nitz, 2015*; *Shine and Wolbers, 2021*; *van Wijngaarden et al., 2020*). These kinds of computations might underlie the emergence of navigational landmarks (*Auger et al., 2012*; *Fischer et al., 2020*; *Jacob et al., 2017*; *Page and Jeffery, 2018*), as sensory stimuli that appear in the egocentric view and, having been deemed reliable, are ultimately mapped to an abstract representation of space (*Barry and Burgess, 2014*; *Bicanski and Burgess, 2016*; *Yan et al., 2021*). Altogether, this evidence strongly implicates RSC in the integration of visual orienting cues and HD. Moreover, RSC is also necessary for path integration-based navigation in darkness (*Elduayen and Save, 2014*), likely by integrating motor (*Yamawaki et al., 2016*) and angular velocity signals (*Hennestad et al., 2021*; *Keshavarzi et al., 2022*) together with the incoming HD from thalamic nuclei and POS, to form a representation of orientation. However, while the coordination between POS and ADn HD representations has been already elucidated across wake and sleep states (*Peyrache et al., 2015*), it is unknown how visually-guided changes in HD representation are coordinated between RSC and ADn. We sought to answer this question by performing simultaneous single unit recording in RSC and ADn in freely moving mice while a visual cue was either rotated around the arena or turned off.

## Results

### Differential HD encoding in RSC and ADn

To monitor single unit activity in RSC and ADn, we implanted independently movable tetrodes assembled in a lightweight microdrive (*Voigts et al., 2020*) targeting the two regions simultaneously in nine mice. We additionally recorded from RSC alone in two mice, and used carbon fiber electrodes to record from ADn in one more mouse (see *Figure 1—figure supplement 1A* for electrolytic lesions in ADn and RSC for each mouse). During the recordings, mice could freely roam in a dark circular arena of 50 cm diameter, inside a large sealed box. The only visual cue was provided by an illuminated subset of LEDs spanning an angle of 20° which formed a wider circle outside and above the arena. Fifty-four percent of the units we recorded under these conditions in ADn met our criterion for HD cells, whereas 12% in RSC did so (*Figure 1B&C* and *Figure 1—figure supplement 2A-C* for more examples of HD units). Our HD cell selection method relied on the amount of directional information, the magnitude of the resultant and the concentration parameter from the tuning curve or the von Mises fit (*LaChance et al., 2022*) to the largest peak for multi-peak units against those obtained from shuffling the spikes (*Figure 1—figure supplement 3A*) and finally on the replication of these in two different segments of a session while the cue is stable (ADn: 11 HD units with 2 peaks, 139 with 1; RSC: 2 HD units with 2 peaks, 71 with 1). RSC displayed a modest HD code, in agreement with previous findings in rats and humans (*Chen et al., 1994b*; *Cho and Sharp, 2001*; *Shine et al., 2016*; *Shine and Wolbers, 2021*): directional information in RSC was lower than in ADn (bits/spike, median: ADn HD = 0.084, n=150; RSC HD = 0.015, n=73; ADn non-HD=0.011, n=122; RSC non-HD=0.006, n=584; Kruskal-Wallis test p<0.0001, p<0.0001 for multiple comparisons, except for ADn nonHD versus RSC HD p>0.05, *Figure 1—figure supplement 3B*), as well as the median resultant length (median: ADn HD = 0.21, n=150; RSC HD = 0.084, n=73; ADn non-HD=0.047, n=122; RSC non-HD=0.041, n=584; Kruskal-Wallis test p<0.0001, p<0.0001 for multiple comparisons, except for ADn versus RSC HD, p=0.0031, and nonHD, p=0.028, *Figure 1—figure supplement 3C*). The differences between these two regions were consistent with a the higher degree of multi-modal selectivity in RSC (*Alexander et al., 2020*; *Alexander and Nitz, 2015*; *Cho and Sharp, 2001*; *Laurens et al., 2019*), where multiple spatial correlates, contexts and states are mixed with and influence HD coding.

### Congruent HD responses to visual cue rotations in ADn and RSC

To challenge the mice's sense of orientation and determine whether ADn and RSC similarly update the HD frame in response to changes to visual stimuli, we instantaneously rotated the LED cue around the arena by either 45° or 90° (*Figure 1D*), in trials ranging from 5 to 40 min. Previous work with cue card rotations indicates that HD cells in ADn rotated their preferred directions coherently (*Yoganarasimha et al., 2006*), but it remains unclear if the same applies to RSC. To further investigate whether RSC HD ensembles stay coherent in our behavioral setting, we calculated the angle offsets between the tuning curves of all unique simultaneous HD pairs before and after cue rotation. As expected, the preferred direction difference between pairs of ADn HD neurons remained rigid after cue rotations (*Figure 1F*, circular correlation = 0.91, n=603 pairs, 10 mice). In RSC HD cells, we observed a smaller yet significant correlation (0.28, n=269 pairs, 9 mice) in the preferred direction difference of RSC HD pairs between before and after cue rotations (*Figure 1G*). Similar results were obtained when all sessions were included (correlation ADn = 0.92, n=1,128 pairs; RSC = 0.73, 1523 pairs), suggesting that RSC HD ensembles maintain a certain degree of coherence in the reference frame.

The reduced rigidity of RSC HD ensembles could emerge from variability in the preferred direction of individual units and/or from the strength of the HD tuning. We quantified the stability in HD tuning as the correlation between the preferred firing directions (PFD) of HD cells calculated from two consecutive segments with the same stable cue orientation. We found that RSC's PFDs were significantly stable although less than ADn HD units (*Figure 1—figure supplement 3F*, ADn left n=150, correlation = 0.93, RSC right n=73, correlation = 0.79, p<0.0001 for both). Furthermore, the strength of HD tuning, quantified as the resultant, did not change between two sample trials from selected significant cue rotation sessions across RSC HD cells (*Figure 1—figure supplement 3G* right, n=71, Wilcoxon Signed-Rank test p=0.79, median before = 0.086, after = 0.088) as well as ADn HD cells (*Figure 1—figure supplement 3G* left, n=128, Wilcoxon Signed-Rank test p=0.23, median before = 0.22, after = 0.23).

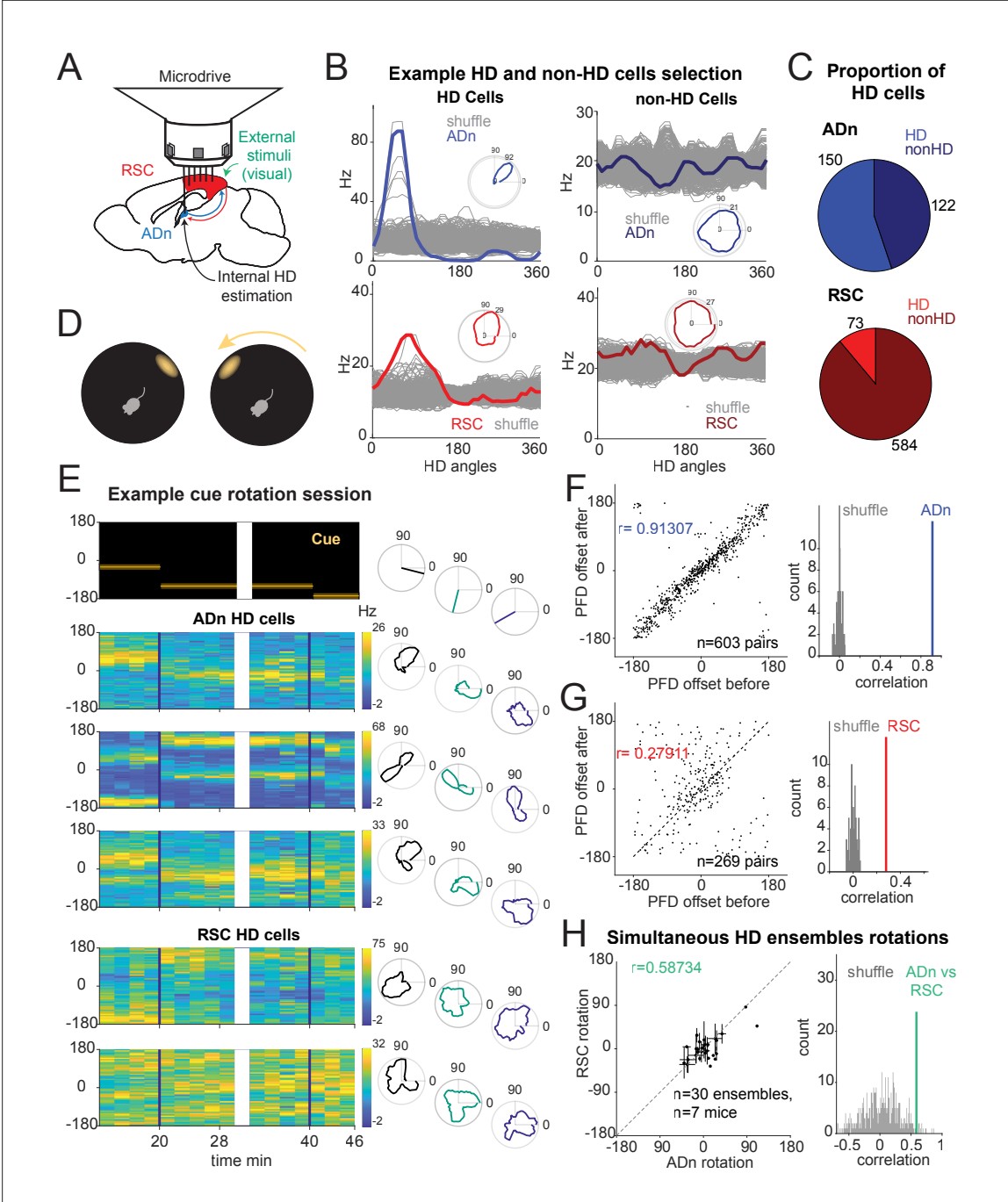

**Figure 1.** Congruent HD response to visual cue rotation in ADn and RSC. (**A**) Schematic of simultaneous ADn (blue) and RSC (red) tetrode recording. (**B**) Tuning curves of examples of HD (left) and non-HD (right) cells in ADn and RSC. Grey lines are the tuning curves obtained from 500 shuffles of the cells firing rates. Insets show the tuning curves in polar coordinates. (**C**) Pie charts showing that 54% of cells in mouse ADn meet the HD selection criterion, but only 12% in RSC do (right) (n=12 mice, 9 with simultaneous ADn and RSC, 2 ADn only and 2 RSC only). (**D**) Schematic of the arena with the only prominent LED cue before (left) and after 90° rotation (right). (**E**) Simultaneous ADn-RSC recording from a session where the cue (top, left) was rotated first by 90° (first segment) and then 45° (second segment). Top right, same cue angular position representation but in polar coordinates. Left, tuning curves (2 min bins) over time of HD cells in the two regions shift the preferred firing direction (yellow bins, maximal firing rate) in response to the cue rotation. Right, same tuning curves in polar coordinates drawn for the different cue angles. (**F**) Right, scatter plot of preferred firing directions (PFD) differences from unique ADn HD cell pairs (n=603) before versus after rotation (ADn correlation: 0.91, 10 mice). Left, the correlation from the data in left is above the 99th percentile of randomly drawn angle differences. (**G**) Same as in F for RSC HD cell pairs (n=269, correlation = 0.28, 9 mice). (**H**) Left, ADn HD ensembles mean rotations (dots with horizontal lines, standard mean error) versus mean rotations of simultaneously recorded HD ensembles

*Figure 1 continued on next page*

*Figure 1 continued*

in RSC (vertical lines, standard mean error). Right, correlation value (correlation = 0.59) from the data in the left (n=30 ensembles for different sizes and directions of rotations from 71 trials, 7 mice) is above the 95th percentile of 500 times shuffled rotation trial indices for each HD cell.

The online version of this article includes the following figure supplement(s) for figure 1:

**Figure supplement 1.** Electrolytic lesions in RSC and ADn.

**Figure supplement 2.** Example HD cells and sorting metrics.

**Figure supplement 3.** HD selection method and HD features across ADn and RSC.

To compare how both regions responded to cue rotations, we calculated the mean preferred direction shifts for RSC and ADn HD ensembles. Despite a bias toward the inertial (self-motion-driven) HD estimate, evident through the higher density of rotations around 0, the rotations from simultaneously recorded ADn and RSC HD ensembles were more correlated (coefficient = 0.54 from unique ensembles for different sizes and directions of rotations, 7 mice) than those produced by random shifting of RSC and ADn neural activity around the cue rotations (*Figure 1H*). The result held when trials from the same ensembles with similar (large or small, negative or positive) ADn rotations were not averaged (*Figure 1—figure supplement 3L*, correlation = 0.48, p<0.01, above the 99th shuffle percentile). This suggested that the HD map is well-locked between the two regions in our behavioral paradigm, despite the differences in HD encoding between ADn and RSC.

## Synchronous shifting of HD representations in ADn and RSC in response to cue rotations

Based on the tuning curves of our HD cells, we concluded that RSC and ADn encode the same HD reference, regardless of whether they shift with cue rotation or disregard the cue as an orienting landmark (*Figure 1H*). However, it was not clear if ADn and RSC also coordinate at a finer temporal scale in response to the cue rotation. To answer this question, we applied a decoding approach to infer the HD representation from the entire recorded population at a high temporal resolution (20ms bins) in ADn and RSC. We implemented a linear-Gaussian generalized model (GLM) that related ADn or RSC ensemble neural activity to HD obtained from behavioral tracking (*Figure 2—figure supplement 1A*). For each trial we estimated the weight coefficients based on the stable cue period before rotation, excluding the last 120 s, and tested the model on the remaining stable period, by calculating the difference between the decoded HD and the HD from headstage tracking, which we will refer to as 'decoded error'. For each session, we compared the accuracy, calculated as the median of the absolute decoded error, with that obtained by decoding HD from shuffled firing rates. For further analyses, we selected sessions where the error was lower than the 10th percentile of 100 shuffles (*Figure 2—figure supplement 1B*). From the remaining sessions, the median decoded error from the test period was 36.32° in ADn (n=36 unique ensembles) and 47.53° in RSC (n=29 ensembles) (p=0.0039, Mann-Whitney test) (*Figure 2—figure supplement 1C and D*). However, given that HD is modulated by angular velocity (*Figure 1—figure supplement 3K*&L, left panels for cue on modulation of the peak firing rate of the HD tuning curves, ADn n=91, median AV >30°/s 53.7 Hz, AV <30/s=50.4 Hz, p<0.0001 Wilcoxon Signed-Rank test; RSC n=27, median AV >30°/s 32.2 Hz, AV <30/s=26.6 Hz, p<0.0001 Wilcoxon Signed-Rank test), we considered whether the decoding accuracy was affected by these state changes. Indeed, decoding performance dropped for high compared to low angular velocity (*Figure 2—figure supplement 1E, F*), (low angular velocity medians: ADn = 33.4°, RSC = 43.4°, high angular velocity medians: ADn = 39.4°, RSC = 60.9°). Two-way ANOVA revealed significant differences between the two regions (p<0.0001) and also between the two velocity states (p=0.0012), but no effect for the interaction between the two variables (p>0.05), suggesting that the angular velocity effect was largely similar between the two regions.

The errors from our GLM decoder reflected the mean rotations of the ensemble HD neurons tuning curves both from ADn and RSC (*Figure 2—figure supplement 1G, H*, 0.73 and 0.75 circular correlation coefficients for ADn, n=113, and RSC, n=59 unique ensembles for different sizes and directions of rotations, respectively, p<0.0001 for both) suggesting that our method largely captured the changes in neural activity with cue rotation. The mean decoded errors after cue rotations from simultaneously recorded ADn and RSC ensembles were also correlated (*Figure 2B*, 0.74 circular correlation coefficient, n=95 unique ensembles for different sizes and directions of rotations, from 8 mice), confirming

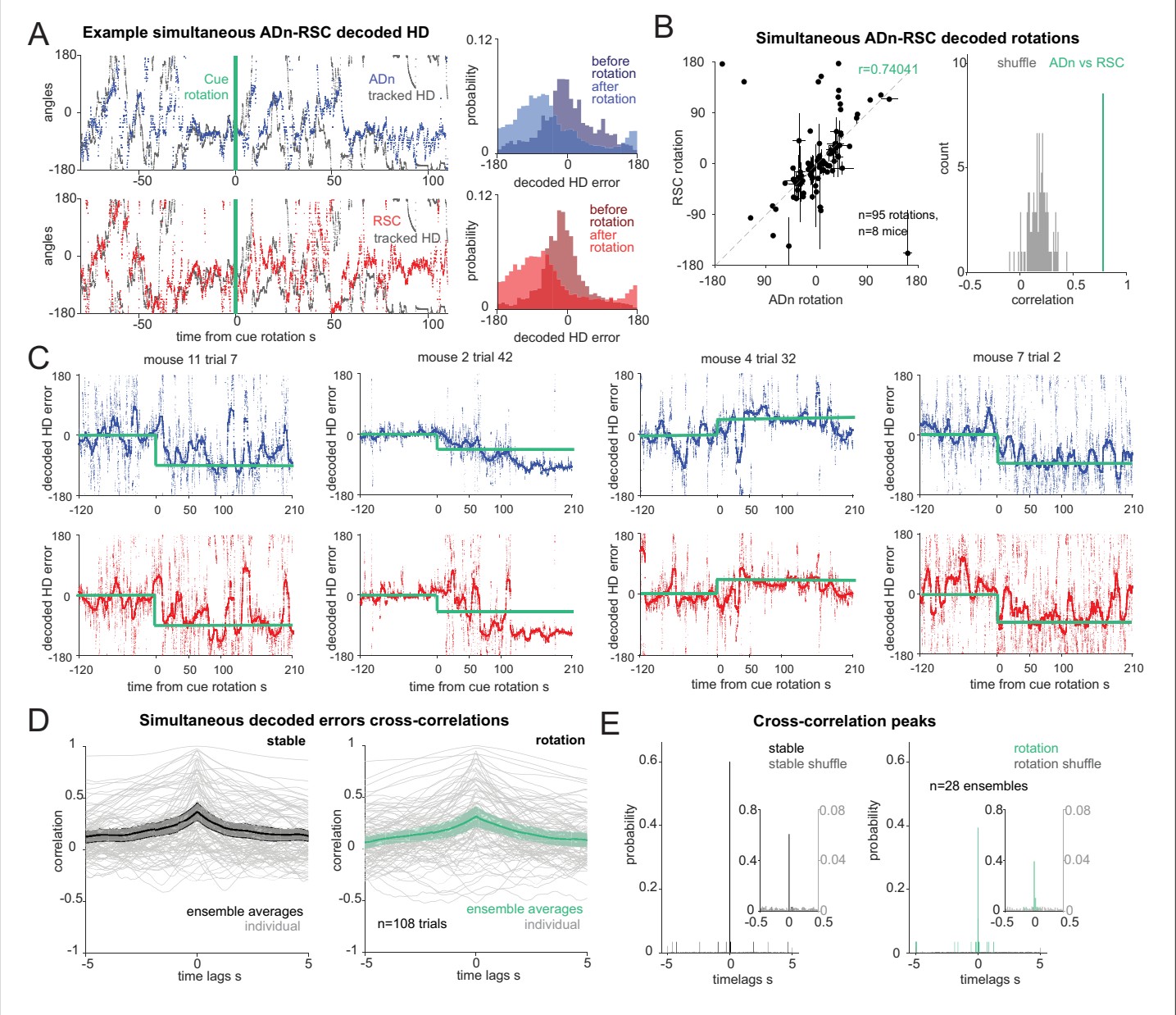

**Figure 2.** Synchronous shifting of ADn and RSC HD representation in response to cue rotation. (**A**) Left, Example of decoded HD in ADn (top, blue line) and RSC (bottom, red line) from a simultaneous recording in the two regions before and after cue rotation (green line at t=0 s). Grey, mouse HD from tracking. Right, probability-histograms of the difference between the tracked and the decoded HD shown on the left in ADn (top, blue) and RSC (bottom, red). Darker shades, decoded error before rotation, lighter shades after rotation. (**B**) Left, decoded ADn vs paired RSC rotation (n=95 unique ensembles for different sizes and directions of rotations from a total of 182 trials, 8 mice). Right, circular correlation coefficient (0.74) between the decoded ADn and RSC rotation (green) is above the 99th percentile of the 100 times randomly shifted RSC decoded HD for each trial (grey histogram). (**C**) Four examples of paired ADn (top row) and RSC (bottom row) decoded HD errors drifting toward the target (yellow). Thicker lines, median-smoothed error over a 5 s window. The fourth example is the decoded error of the traces shown in A. (**D**) Mean and 95% CI of temporal cross correlation between paired decoded errors before rotation (left, black) and immediately after rotation (right, green) (75 s long segments, n=28 unique ensembles from a total of 108 paired trials out of 182 with mean ADn rotation >17.2°, 8 mice). Grey, individual trials. (**E**) Probability histograms (20ms bins) of the time lags corresponding to the peak correlation values from the unique ensembles (n=28), averaged traces from the trials in D; left, before rotation, right, after rotation. Left y-axis scaled to show the uniformity of the null distributions (grey). Insets, zoomed in histograms in the –0.5 s to 0.5 s range. The real distributions are significantly different from null (two-sample Kolmogorov-Smirnov test, p<0.0001 for both stable and shifted). No difference between stable and shifted trial correlations was observed (Wilcoxon Signed-Rank test p=0.88).

The online version of this article includes the following figure supplement(s) for figure 2:

**Figure supplement 1.** HD decoding with a linear-Gaussian GLM.

*Figure 2 continued on next page*

*Figure 2 continued*

**Figure supplement 2.** Relationship between cue bearing and decoded neural rotations.

**Figure supplement 3.** Robustness of different similarity metrics between decoded ADn and RSC HD and time course of the realignment.

the findings from the mean tuning curves rotations (*Figure 1H*). Similarly, we also observed a high density of decoded rotation values around zero. This effect likely resulted from devaluation or bias toward internal HD estimates caused by the familiarity with the arena and by the consecutive cue change creating a mismatch between the internal HD and the visually HD reference (*Knierim et al., 1998*).

To investigate whether the egocentric experience of the cue influenced the rate of under-rotations, we compared rotations occurring when the cue was well outside of the visual field of the mouse before rotation (*Dräger and Olsen, 1980*; *Sterratt et al., 2013*) or not (>154° or >-154°, calculated at the center of the cue, with 0° aligned to the mouse' snout; *Figure 2—figure supplement 2A*). We found that, despite an overall effect of the size of the cue rotation which led to on-average better cue control for small rotations than for big rotations (p=0.0008 ANOVA with three factors, with p>0.05 as the effect of region and view category and all interactions), there were no differences between the individual groups, (p>0.05 multiple comparison after Bonferroni correction, n cue out of view: small rotation 14 trials, big rotation 8 trials; cue in view: small rotation 36 trials, big rotation 51 trials, *Figure 2—figure supplement 2B*). This suggested that even though large mismatches were more likely to result in under-rotations, we could pool the results of the rotation analyses. Furthermore, we did not find any correlation between the size of the decoded rotation in ADn and RSC and the egocentric bearing of the cue before (*Figure 2—figure supplement 2C*, –0.034 and –0.031 correlation coefficients respectively for ADn, n=253, and RSC, n=262) or after rotation (*Figure 2—figure supplement 2D*, 0.007 and –0.023 Pearson correlation coefficients respectively for ADn and RSC). Altogether, these analyses indicate that, in our behavioral setting, the initial egocentric experience of the cue rotation was not a factor in the under-rotations of the HD representation.

Next, we asked how changes in environmental stimuli alter the rate of HD reference shifts, and if these differ across brain regions. By applying our decoding strategy, we first observed that decoded errors drifted to a new HD reference at a variable speed (*Figure 2C*). On average we observed similar, and mostly slow, HD reference shifts in both ADn and RSC (*Figure 2—figure supplement 3H*). Interestingly, we observed closely matched trajectories to the new HD offsets in simultaneously recorded neurons from both ADn and RSC (*Figure 2C*). To quantify if initial jitter in the drift of the HD representations were indicative of a region-specific response to the new angular position of the cue, we calculated the temporal cross-correlations between simultaneous ADn and RSC decoded errors immediately after cue rotations (75 s window). To isolate the effect of change of the HD reference, we included only trials where the absolute mean HD shift was at least 17.2° or above. While some trial-to-trial variability in the correlation level was observed, the majority of trials had peaks at 0ms time lags (two-sample Kolmogorov-Smirnov test between data and null distribution, p<0.0001 for both stable and shifted), both before and after cue rotations (*Figure 2D and E*; Wilcoxon Signed-Rank test p=0.88, n=28 unique ensembles from 108 trials). Similar results were obtained for shorter decoded error traces immediately after cue rotation (25 s window), albeit with higher variability (*Figure 2—figure supplement 3A*, two-sample Kolmogorov-Smirnov p<0.0001 for both stable and shifted; Wilcoxon Signed-Rank test between shifted and stable p=0.40). This confirmed that the two regions were coordinated even in the initial stages of the shifting of the HD representation.

To check that the 0 lag peaks were not originating from spurious cross-correlation values of long and jittering decoded error traces, we assessed the difference in decoded HD and decoded HD error between the two regions, in short (5 s) segments to cover a total of 35 s and plotted the absolute median for different time lags. In principle, this approach should mirror that of the temporal cross-correlation, whereby the time lags corresponding to the smallest difference values would reveal whether there was a temporal offset between the two HD representations. The difference in the decoded HD showed a trend toward 0ms coordination (*Figure 2—figure supplement 3B*, ensemble averages from 108 individual trials), but this effect was weak. In theory, if there was a detectable temporal offset, it would be masked by opposing HD difference values caused by varying AV directions and directions of rotations. To obviate these problems, we evaluated the difference in decoded HD errors (*Figure 2—figure supplement 3C*), after ensuring the final rotations had all the same sign.

We found that a great majority of the trials had troughs at 0ms lag (*Figure 2—figure supplement 3D*, n=108 trials, Wilcoxon Signed-Rank test p=0.83). We evaluated this relationship for independent ensembles (n=28) by taking the median across trials from the same ensembles (*Figure 2—figure supplement 3E, F*) and found the same result (Wilcoxon Signed-Rank test p=0.92). We ensured that the variance in the –5–5 s lags was correlated with the variance of the tracked HD (*Figure 2—figure supplement 3G*, n=108 trials, correlation *r*=0.389 for the stable, *r*=0.302 after rotation, p<0.0001 for both).

Contrary to our predictions (i.e. that RSC would lead the HD reference update via visual integration of the cue's new angular position), we found that the two regions were synchronized (within 20ms) in this regard. Variability was observed in the histograms from the cross-correlations after cue rotation, but they lacked any specific bias for anticipatory or delayed time lags. Changes in the neural activity, and therefore in HD decoding, could emerge following the cue rotation as a response to visual stimuli, potentially explaining the slightly decreased synchrony compared to stable cue periods (*Figure 2E*).

## Correlated HD drift in darkness in ADn and RSC

Visual cues are essential for stabilizing the HD reference map and guide orientation, however it is unknown if they are necessary for maintaining the coordination of HD representations in ADn and RSC. To resolve this issue, we challenged the sense of orientation in a subset of mice by turning the LED off after a period of cue-on baseline (*Figure 3A*). HD cells maintained the same initial preferred directions while the cue was on in a stable position, but were more variable while the cue was off, and sometimes continuously drifted during prolonged darkness (*Figure 3B*). The mean firing rates of the recorded ensembles were stable between cue-on and -off conditions in ADn (n=155, p=0.067 Wilcoxon Signed-Rank test) and marginally reduced in RSC (n=159, p=0.046), suggesting a small response of cortex to the change in light conditions (*Figure 1—figure supplement 3H*). To quantify the effect on HD coding properties, we evaluated the tuning curves every 2 min in the cue-on and cue-off periods, to avoid combining long HD drifts. The peak firing rates of HD cells both in ADn and RSC were reduced (*Figure 1—figure supplement 3I*, ADn n=91 median cue-on 62.52 Hz, cue-off 57.63 Hz, p=0.0027, RSC n=27 median cue-on 41.36 Hz, cue-off 37.74 Hz, p=0.0004, Wilcoxon Signed-Rank test), but only RSC's HD resultant lengths were affected by darkness (*Figure 1—figure supplement 3J*, cue-on median 0.123, cue-off median 0.087, p=0.0057 for RSC, cue-on median 0.22, cue-off median 0.21, p=0.69 for ADn), consistent with previous reports of varied HD responses to different light conditions in RSC (*Chen et al., 1994a*).

On average, we observed modest levels of drift in darkness (*Figure 3F*, n=15 ADn ensembles from 58 trials, p=0.0053 Wilcoxon Signed-Rank test between cue-on drift, median = 15.9°, and cue-off drift, median = 25.6°). When we applied the same decoding strategy as in the rotation trials to quantify the drift in the two regions, we observed correlated HD representations between ADn and RSC both during cue-on and cue-off periods (*Figure 3C* and *Figure 3—figure supplement 1C*). This is evidenced by the diagonal band across the column-normalized 2D histograms of the HD decoded errors averaged across the different ensembles (*Figure 3D&E*, left). The circular correlations of the decoded drifts were similar between cue-on (*r*=0.277) and cue-off periods (*r*=0.256) (p=0.677, Wilcoxon Signed-Rank test, n=12 unique ensembles) and were significantly higher than the correlations with shuffled RSC drifts (*Figure 3D&E*, right), suggesting that in the two light conditions the HD representations of the two regions stayed coordinated.

In the awake behaving rodent, angular velocity drives the update of the ongoing HD, which, in the absence of prominent visual cues, can drift inconsistently from the current HD reference (*Skaggs et al., 1995*; *Stackman et al., 2002*; *Valerio and Taube, 2016*). Moreover, it modulates ADn and RSC peak firing of HD neurons in both cue-on and cue-off conditions (p<0.0001 between AV >30°/s and AV <30°/s for both cue-on medians = 53.7 and 50.4, respectively, and cue-off medians = 51.2 and 47.2 for ADn, n=91; p<0.0001 also between AV >30°/s and AV <30°/s for both cue-on medians = 32.2 and 26.6, respectively, and cue-off medians = 30.8 and 26.3 for RSC, n=27, Wilcoxon Signed-Rank tests between AV states). We next asked how this variable affects the relationship between the internal HD of ADn and RSC when the cue is turned off. We isolated time points of high and low angular velocity (cut-off of 30°/s) and calculated the circular correlation between ADn and RSC HD drifts in the two light conditions. We found that the coordination between the two regions is similar between cue-on and cue-off conditions for low AV and slightly reduced during cue-off for high AV (*Figure 3—figure*

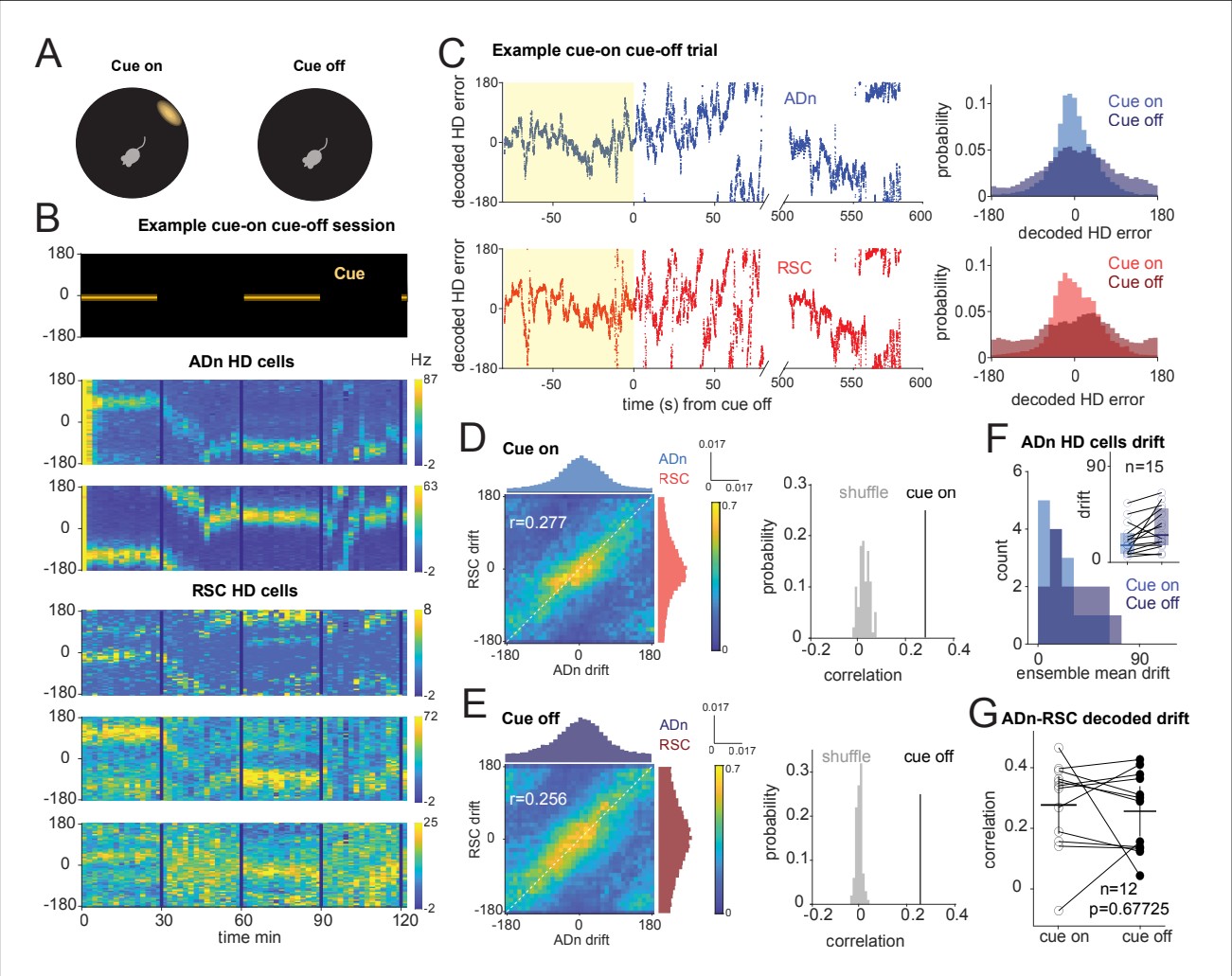

**Figure 3.** Correlated HD drift in darkness in ADn and RSC. (**A**) Schematic of cue-on/off trials. (**B**) Simultaneous ADn-RSC recording from a session where the cue (top) was turned on and off. (**C**) Left, simultaneous ADn (blue) and RSC (red) decoded HD errors from the first cue-on (yellow shaded) and -off trial of the example shown in B. Black, median-smoothed decoded error over a 5 s window. Right, probability normalized histograms of the decoded HD error from the example in C in ADn (blue, top) and RSC (red, bottom). The lighter shaded histograms are from the cue-on segments, the darker shaded from the cue-off. (**D**) Left: 2D histogram of simultaneous ADn and RSC decoded errors from cue-on, normalized by the maximum column value per bin (i.e. in the ADn dimension); above and on the side, marginal distributions of ADn and RSC drift, respectively. Right: correlation value (*r*=0.277) from the not-normalized data in the left is above the 99th percentile of the distribution obtained after randomly shifting the RSC drift in each trial (grey). (**E**) Left: same 2-D histogram and marginal distributions for the cue-off segments from the same trials as D; right: averaged circular correlation (*r*=0.256) of the real cue off data is also above the 99th percentile of the shuffle distribution. Data in D and F includes n=12 ensembles averaged from 28 trials, 3 mice. (**F**) Distribution of the absolute drift of ensemble ADn HD tuning curves averaged over 2 min bins and across unique ensembles (n=15 from 58 trials, p=0.0053 Wilcoxon Signed-Rank test between cue on drift, inset individual values, median = 15.9, and cue off drift, median = 25.6, shaded areas are the 95% CI). (**G**) The circular correlation between decoded ADn and RSC drifts from unique ensembles is not significantly different between cue on and cue off (n=12, p=0.67, mean correlations 0.277 for cue on and 0.256 for cue off, plotted together with the 95% CI).

The online version of this article includes the following figure supplement(s) for figure 3:

**Figure supplement 1.** Effect of angular velocity (AV) on HD drift in darkness in ADn and RSC.

*supplement 1A*, low AV cue-on *r*=0.262 vs cue-off *r*=0.300 p=0.38, high AV cue-on *r*=0.256 versus cue-off 0.177, p=0.042 Wilcoxon Signed-Rank test), where larger decoded errors have already been observed (*Figure 2—figure supplement 1E, F*). Our results therefore indicate that the HD representation is also coordinated between ADn and RSC in the absence of a visual input, but can be challenged during periods of HD instability.

## Dense HD signal in the ADn-to-RSC connectivity

Temporal coordination of HD representation on the order of 20ms or less across different structures could be accomplished by direct monosynaptic connections or by concurrent input from different areas, particularly for the visual update of the HD reference. To investigate the anatomical substrate for direct connectivity between ADn and RSC, we performed retrograde monosynaptic rabies tracing (*Wickersham et al., 2007*) experiments in ADn and RSC. We found that RSC cells from the same areas where we performed tetrode recordings (*Figure 1—figure supplement 1A and B*) received dense anterior thalamic inputs, and particularly ADn (*Figure 4A*, *Figure 4—figure supplement 1A*). Conversely, ADn exhibited surprisingly sparse presynaptic RSC cell labeling. These cells were frequently localized in the granular and ventral portion of RSC (*Figure 4B*). We further assessed ADn's presynaptic labeling with an alternative method, to circumvent potential biases in the viral trans-synaptic labeling and poor tropism (*Rogers and Beier, 2021*). Retrobeads injected in ADn labeled more densely the deep layer 6 of RSC, in line with the canonical corticothalamic circuit architecture (*Harris et al., 2019*), but again, more specifically the granular portion. Overall, these results are consistent with previous studies in the rat (*Shibata, 1998*; *Shibata, 1993*; *van Groen and Wyss, 1990a*; *Vantomme et al., 2020*).

We next examined whether the HD coordination we observed in our recordings could emerge from ensembles in RSC or ADn that were functionally connected and could convey updated or visually anchoring HD information. To this end, we performed spike cross correlation between all possible ADn-RSC pairs, considering the spikes occurring during cue-on periods. Putative monosynaptic connections were identified in the cross correlograms as sharp peaks above the baseline (*Figure 4C*) between 1 and 5ms from the time of ADn unit firing (*Figure 4E*). Using this metric (*Peyrache et al., 2015*; *Stark and Abeles, 2009*), we identified 6.88% of all possible pairs with a monosynaptic connection in the ADn-to-RSC direction (348 out of 5056) and only 0.08% in the RSC-to-ADn direction (4 out of 5056). In terms of unit counts, 91 out of 326 ADn units had at least one connection to RSC, while only 4 out of 749 RSC units had a connection with ADn (*Figure 4D*). The mean number of RSC synaptic partners was 4.23 for ADn HD cells and 0.04 for non-HD cells. When we focused on the connectivity during darkness, we found similar results: 35 out of 161 units in ADn (3 mice) had at a connection, while no RSC units were connected. The generally lower connectivity rate was likely due to the reduced sample size of cue-off trials, but suggested that even in the absence of visual cues corticothalamic projections were not a substrate for HD coordination. ADn's dense connectivity largely originated from identified HD cells (275 connected pairs from ADn HD cells versus 72 from non-HD ADn cells) and included both HD as well as non-HD partner cells in RSC (*Figure 4F*). HD-coding was a feature of both RS and FS units in RSC (*Figure 4—figure supplement 2*), and both groups received ADn connections (*Figure 4G*, left plot).

Finally, the distribution of the preferred direction differences between ADn HD-coding units and their HD-coding synaptic partners in RSC showed a small but significant bias toward similar tuning (*Figure 4G* left polar plot, n=73 units, 28 RS and 47 FS, 6 mice, circular mean –5.4° with a circular standard deviation of 61°, Rayleigh test for non-uniformity p=0.005 for the RS synaptic partners only, p=0.002 when both cell types were combined). This similarity bias was not observed for the preferred direction differences between the same ADn units and all other non-synaptically connected HD-coding units, whose distribution was uniform (*Figure 4G* right polar plot, n=73, 65 RS and 8 FS units, Rayleigh test p=0.052 for the RS units, p=0.086 for RS and FS combined). Together, these results suggest that ADn sustains the RSC HD code with a widespread feedforward connectivity to both RS and FS units, a connectivity that targets not only HD-tuned units, directly shaping their preferred directions (*Figure 4H*), but also units with more complex, presumably multimodal, receptive fields. On the other hand, the very sparse RSC-to-ADn connections that we identified are, alone, unlikely to drive the change or the stability, in the presence of visual cues, in preferred directions in ADn (*Figure 4I*).

## Discussion

Our data show that the HD representation in ADn and RSC is closely coordinated, both in conditions when visual cues are stable and during adaptation to a new reference in response to cue rotation (*Figure 2D and E*). This is also true for HD drift in darkness (*Figure 3D, E and G*), showing that visual input is not necessary for maintaining these coordinated representations, and that other sensory

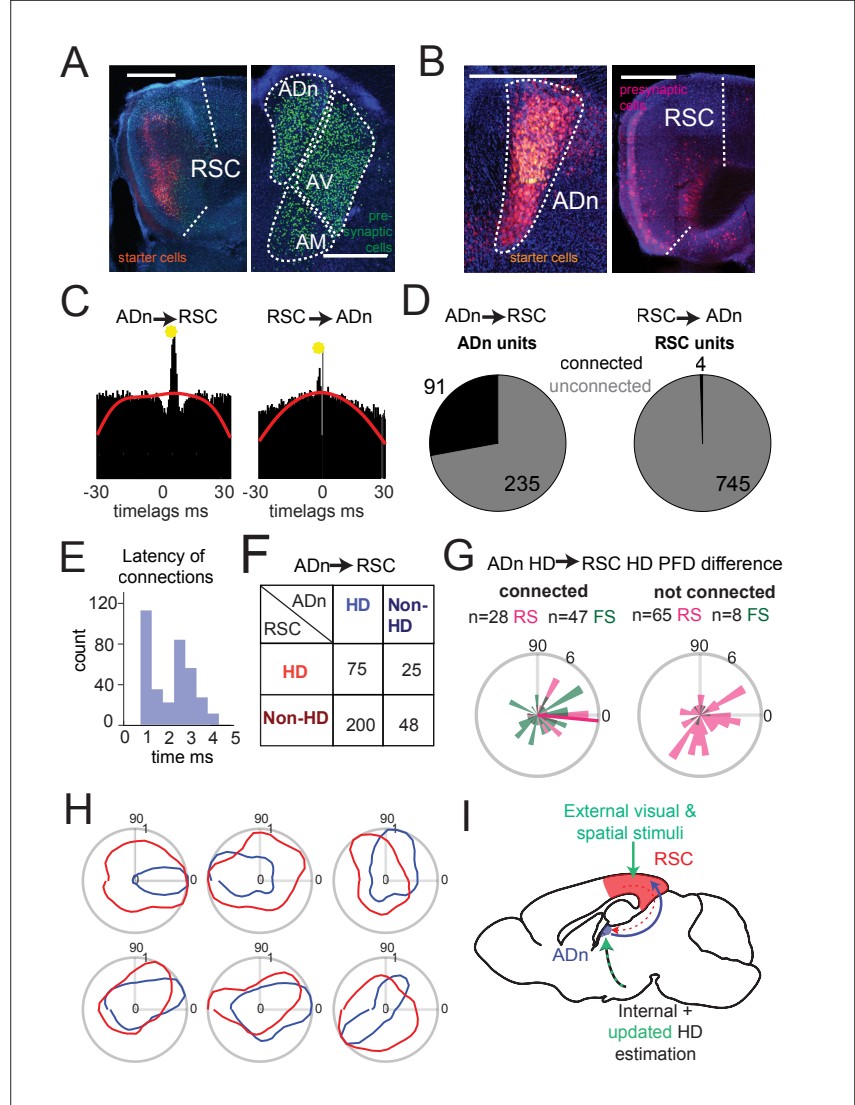

**Figure 4.** Dense connectivity from ADn to RSC. (**A**) Monosynaptic rabies tracing of inputs to RSC (left, starter cells in red) shows a high density of presynaptic cells in ADn (right, green). (**B**) Monosynaptic rabies tracing of inputs to ADn (left, starter cells identified by the overlap of blue, green and red) shows a low density of presynaptic cells in RSC (right, red) and mostly in A29. Scale bar in A and B, 0.5 mm. (**C**) Examples of cross-correlograms with putative excitatory connections (yellow circle) from ADn to RSC (left) and RSC to ADn (right), showing a sharp peak between 1 and 5ms time lag above the baseline (red line) at more than 99.9% of the cumulative Poisson distribution. (**D**) Number of ADn units with putative connections to RSC (left) and of RSC units with putative connections to ADn (right). (**E**) Distribution of the latencies of the peaks in the cross-correlograms for ADn-to-RSC connections. (**F**) Breakdown into HD- and non-HD coding of the putative pre- and post- synaptic partners of connected ADn-to-RSC pairs. (**G**) Polar plot distributions of the differences between preferred directions of connected ADn HD units and their putative HD-coding synaptic partners in RSC (n=75 units, 28 RS and 47 FS, 6 mice, left) and the same ADn HD units and all other HD-coding non-synaptic partners (n=73, 65 RS and 8 FS, 6 mice, right). Magenta line on the left plot indicates the circular mean (–5.4°) of the RS peak differences; Rayleigh test for non-uniformity p=0.005 for the RS synaptic partner, p=0.002 for RS and FS, p=0.052 of the RS non-synaptic partners and 0.086 for all non-partners (right plot). (**H**) Polar plots of tuning curves of 6 example pairs of connected ADn to RSC HD units with variable preferred directions. (**I**) Adapted schematic from *Figure 1A* showing that the connectivity from RSC to ADn is nearly absent and that the visually-guided updates in the HD frame emerge from a strong feedforward HD input from ADn.

The online version of this article includes the following figure supplement(s) for figure 4:

**Figure supplement 1.** Anatomical tracing of reciprocal connectivity between ADn and RSC.

**Figure supplement 2.** Separation of putative pyramidal neurons and fast-spiking interneurons in cortex.

modalities, such as angular velocity, optic flow, and/or motor efference copy, influence ADn and RSC HD. Finally, our functional connectivity (*Figure 4D*) supports the ADn-RSC HD coordination through a strong feedforward input to RSC, which shapes the local HD code there (*Figure 4F–H*). Specifically, the bias toward similar tuning between connected ADn and RSC HD units suggests that ADn HD code might not be simply inherited in RSC, as has been shown in POS (*Peyrache et al., 2015*; *Peyrache et al., 2017*), but likely integrated with other spatial codes via different circuit organization principles, that include recruitment of FS and RS neurons (*Simonnet et al., 2017*). We conclude that the visually driven updating of the internal HD is a more complex process that privileges coordination across brain regions over sustained error signals with mismatched representations. However, the temporal resolution of the decoding occludes potential faster dynamics, at the scale of monosynaptic connections (ms range). Furthermore, given the reports of cells that track internal and external spaces and are able to bind HD and prominent visual information especially in dysgranular RSC (*Jacob et al., 2017*; *Sit and Goard, 2023*) and in posterior cortices (*LaChance et al., 2022*), further investigation is needed on the presence of these specific firing patterns in set ups similar to that described here and on their activity in relation to ADn firing to better understand the mechanisms of realignment.

Using a simple generalized linear encoding model of HD (*Figure 2—figure supplement 1A*), we decoded HD at a fine (20ms) temporal scale with ensembles of 3–30 units ADn and 8–110 in RSC. We observed variable and mostly slow drifts of the HD representation to a new target (*Figure 2C* and *Figure 2—figure supplement 3H*). Repeated exposure to an unstable cue is known to cause landmark devaluation (*Knierim et al., 1995*; *Knight and Hayman, 2014*) and, together with extended navigation within the same environment, increased the incidence of under- or 0° rotations in our data (*Figure 2—figure supplement 2B*). Whether these conditions affect the speed of these shifts in our paradigm is unknown. Slow, continuous drifts of the HD representation after cue rotation as reported here (*Figure 2C*, *Figure 2—figure supplement 3H*) have also been observed in flies (*Kim, 2017*) and in rat (*Knierim et al., 1998*) and mouse (*Ajabi et al., 2023*) ADn, with the exception of one study in the rat ADn (*Zugaro et al., 2003*) where immediate HD shifting was observed in specific cue-heading configurations. While behavioral and arena-configuration differences may underlie this discrepancy, it remains to be resolved if the cause of such slow drifts lies in: (1) the configuration of the polarizing visual stimulus, including the distance between the cue and the mouse, with farther away cues being less impacted by egocentric view and having more control over HD *Zugaro et al., 2001*; (2) the memory of previous experiences of cue rotations (*Ajabi et al., 2023*; *Knierim et al., 1995*); and/or (3) the intrinsic time course of synaptic plasticity associated with the learning of the new landmark orientation (*Goodridge et al., 1998*; *Kim et al., 2019*; *Kim, 2017*; *Page et al., 2014*; *Skaggs et al., 1995*; *Yan et al., 2021*).

Whether distinct circuit mechanisms are recruited to coordinate the learning of the new orienting cue according to the behavioral demands and the complexity of the navigation task is not known. POS, through its reciprocal connections with visual areas, could provide the visual reference information to ADn, RSC and LMN, the obligatory HD path upstream of ADn (*Yoder et al., 2015*; *Yoder and Taube, 2011*). Another possible route includes the cortico-thalamic control through thalamic reticular nucleus, which readily and densely inhibits ADn and receives presubicular and retrosplenial connections (*Vantomme et al., 2020*).

Our functional connectivity experiments reveal a strong feedforward ADn-to-RSC HD drive (*Figure 4D–G*) and sparse RSC-to-ADn connections. This asymmetry was more extreme than that observed in previously reported ADn-POS connectivity (*Peyrache et al., 2015*; *van Groen and Wyss, 1990b*), and possibly exacerbated by the widespread sampling of RSC locations in our recordings, across granular and dysgranular portions, with a higher concentration of tetrodes in the upper L2/3 and L5 (*Figure 1—figure supplement 1*) and 9 out of 48 cortical tetrodes in mice with simultaneous ADn and RSC recordings in L6. Further denser sampling of layer 6 in RSC, and more specifically in the granular portion, where ADn's presynaptic partners in RSC are mostly found (*Shibata, 1998*; *Figure 4B*, *Figure 4—figure supplement 1B*), are needed to clarify the role of the anatomically-identified corticothalamic projections in the visual updating of ADn-RSC HD reference. RSC also extends over 1.5 mm in the anterior posterior axis, and it is possible our recordings have missed hotspots in the more posterior granular RSC bordering POS. Given that RSC displays a gradient along the anterior-posterior axis in encoding egocentric versus allocentric navigational variables (*Hennestad*

*et al., 2021*), future studies should address whether this anterior-posterior specialization is reflected not only in visual and HD computations, but also on downstream synaptic connectivity.

How would the visual cue integration that anchors HD be reflected in the ensemble representation? We hypothesized that an 'error' signal would appear as a temporal offset in the HD of the two regions: specifically, the RSC HD update by integration of visual inputs, would precede that of other regions, in our case ADn. Contrary to this hypothesis, our decoding showed no temporal offset in HD representation during shifting (*Figure 2D and E* and *Figure 2—figure supplement 3A–F*). Importantly, this was true regardless of the exact control of the cue over the realignment. At the same time, even in the absence of visual cues, RSC and ADn were closely coordinated during small and large HD drifts, a phenomenon that has recently been proposed to be mediated by intact cerebellar inputs (*Fallahnezhad et al., 2023*). This coordination, which we directly show under two visual challenges, is likely sustained by a strong and widespread feedforward ADn-to-RSC connectivity, where the updated HD reference may already be computed upstream of ADn (*Yoder et al., 2015*). This framework is consistent with an existing hypothesis that visual anchoring may compete with and, depending on the manipulation, dynamically bias the memory of the internal estimation from angular velocity (*Ajabi et al., 2023*; *Knierim et al., 1998*).

In conclusion, our study provides new insight on the relative dynamics of HD realignment and drift and on the direct connectivity between ADn and RSC ensembles. The HD coordination and striking sparseness of RSC-to-ADn connectivity do not preclude, however, that RSC could support the change in HD reference through activation of dedicated ensembles encoding the orienting 'landmark' (*Bicanski and Burgess, 2016*; *Mitchell et al., 2018*; *Page and Jeffery, 2018*) or with conjunctive HD-visual fields. In fact, RSC is necessary for ADn HD alignment to visual cues (*Clark et al., 2010*) and the dense interconnection between RSC and several regions of the hippocampal formation (*Sugar et al., 2011*; *Wyss and Van Groen, 1992*) may support coordinated HD representation across the brain as a mechanism to ensure consistent flexible spatial computations relevant to behavior output. Future experiments using multi-site high-density recordings with laminar probes in RSC could directly assess the activity patterns, at the single unit and population level, associated with the learning of the new cue orientation and the reliability (or unreliability) of a landmark.

## Methods

### Behavior and subjects

All animal procedures were performed in accordance with NIH and Massachusetts Institute of Technology Committee on Animal care guidelines. We used adult (>8 weeks old) C57BL/6 from Charles River and from Jackson Laboratory RRID: IMSR_JAX:000664 and one Vgat-Ires-Cre C57 BL/6 mice (RRID:IMSR_RBRC10723). Four females and eight males were used for tetrode recordings, and two 12-week-old mice were used for modified rabies tracing experiments in RSC, one mouse for rabies and one for red retrobeads tracing experiments in ADn. Mice were kept on a 12-hr light/dark cycle with unrestricted access to water. Eight of the implanted mice underwent mild (up to a 10% reduction in body weight) food restriction. Of the implanted mice, eight were housed isolated in conventional cages, four with siblings in rat cages with running wheels. One mouse had channelrhodopsin expression in cortical interneurons, but this aspect was not investigated in the present study.

The behavioral arena was 50 cm in diameter with a 25 cm cylinder wall, surrounded by an outer cylinder of 80 cm diameter and 30 cm height, where a string of 132 white LEDs (Adafruit, APA102) covering the upper circumference provided the only light source. The arena was enclosed in a 78x86 × 84 cm wooden dark box to shield from lighting and noise. In food deprived mice, pellets (Bioserve) were sprinkled on the floor to allow continuous exploration during long recordings. To provide novelty in the environment and induce exploration, two types of arena walls (black pvc with a white paper at the upper edge and opaque clear plastic) were used and changed when the cue rotation did not produce shifts in the HD tuning.

The visual cue was a set of computer-controlled (Teensy 3.2) LEDs spanning an angle of 20° with brightness following a gaussian with peak at the center and sd of 1. 2 weeks after surgery mice were habituated to a single cue or no cue at all while units were monitored. Different starting cue angular positions for the recording sessions were sampled and different sequences of rotations of ±90° and ±45° were played. For recordings with the Open-Ephys ONIX system (*Newman et al., 2023*) with

the commutator, rotations and cue on-off switches occurred every 20–40 min, versus the 5–20 min for recordings with the first generation Open-Ephys system (*Siegle et al., 2017*) without commutator. Sessions length varied based on the animals' behavior, with a minimum of 2 up to 11 rotations/on-off switches. Cue rotations occurred in consecutive 'jumps' from one angular position to the next. In three mice, periods of darkness were interleaved in some rotation sessions.

## Electrodes and drive implants surgeries

Light weight drives for tetrode recordings were fabricated following the guidelines in *Voigts et al., 2020* for a total of 16 independently movable tetrodes per drive. Arrays were designed to simultaneously target ipsilateral ADn and RSC, for a total length of 2.8 mm and a width of 0.5 mm. To increase the yield of units especially for ADn, some guide tube positions were occupied by two tetrodes. Tetrodes were constructed from 12.7 μm nichrome wired (Sandvik – Kanthal, QH PAC polyimide coated) with an automated tetrode twisting machine (*Newman et al., 2020*) and were gold plated to lower the impedance to a final value between 150 and 300 kOhm. One mouse was implanted with 32 carbon fiber electrodes ~100 Ohm (*Guitchounts et al., 2013*) in ADn only, whose position was fixed since surgery.

All surgeries were performed using aseptic techniques. Mice were anesthetized with isoflurane (2% induction, 0.75–1.25% maintenance in 1 l/min oxygen) and secured in a stereotaxic apparatus. Body temperature was maintained with a feedback-controlled heating pad (DC Temperature Control System, FHC). Slow-release buprenorphine (1 mg/kg) and dexamethasone (4 mg/kg) were pre-operatively injected subcutaneously. After shaving of the scalp, application of hair-removal cream and disinfection with iodine and ethanol, an incision was made to expose the skull. For implants, after cleaning with ethanol, the skull was scored and a base of dental cement (C&B Metabond and Ivoclar Vivadent Tetric EvoFlow) was applied. A burr hole was drilled over prefrontal cortex close to the olfactory bulb for placement of the ground screw (stainless steel) connected to a silver wire. Sometimes an additional burr hole and ground screw, connected to the other with silver epoxy, provided extra stability. For drive implants with tetrode arrays, a large craniotomy from ~0.3 to~3 mm from Bregma, and from the midline to ~0.95 mm ML at the level of M2 and ~0.7 mm ML at the level of RSC was drilled. After durotomy, the drive was lowered onto the surface of the brain with one RSC (AP ~2.400, ML ~0.150 mm, DV ~0.200 mm) and one ADn (AP ~0.350 mm, ML ~0.975 mm, DV ~1.800 mm) -targeting tetrodes extended for guiding the placement of the array. For the carbon fibers implant, a smaller (~1 mm diameter) craniotomy, followed by durotomy, allowed lowering of the bundle of fibers into ADn (AP: 0.68 mm, ML 0.75 mm, DV 2.65 mm). The drive, or the fiber frame, was then secured to the skull with dental cement, the skin incision was partially closed with sutures and the mouse was placed in a clean cage with wet food and a heating pad and monitored until fully recovered. All drive implants were done on the right hemisphere.

## Viral and retrobead surgeries

The same stereotactic procedures were applied to viral and beads surgeries for anatomical tracing. For ADn rabies tracing (*Figure 4B*), a burr hole was drilled over AP 0.68 mm, ML 0.75 mm coordinates and 25 nL of 1:1:1 mixture of helper viruses, pAAV-syn-FLEX-splitTVA-EGFP-tTA and pAAV-TREtight-mTagBFP2-B19G (Wickersham) and AAV2/1.hSyn.Cre (Janelia Farms) was delivered at a rate of 60 nL/min through a glass pipette lowered to DV 2.65 mm. This injection was followed by 50 nL of (EnvA) SAD-ΔG-mCherry (Wickersham) two weeks later at the same location, and after 7 days the brains were processed for histology. For RSC rabies tracing experiments (*Figure 4A* and *Figure 4—figure supplement 1A*), the coordinates were AP 2.8 mm, ML 0.45 mm, DV 0.75 and 0.45 mm, and the injection of 50 nL of 1:1 mixture of AAV2/1.hSyn.Cre (Janelia Farms) and AAV1-hsyn-DIO-TVA66T-dTom-CVS-N2C(g) (Allen Institute) was followed 3 weeks later by a 100 nL of EnvA dG CVS-N2C Histone-eGFP (Allen Institute) before histological processing 9 days later. In an additional tracing experiment (*Figure 4—figure supplement 1B*), following the same surgical procedures, 20 nL of red fluorescent microspheres (Lumafluor) were injected into ADn at AP 0.78 mm, ML 0.74 mm coordinates and DV 2.788 mm and the brain was processed for histology 7 days after the injection. Five min after each injection, the pipette was slowly withdrawn and the incision was sutured.

## Immunohistochemistry and confocal imaging

Brain fixation with 4% paraformaldehyde in PBS was achieved with transcardial perfusion for monosynaptic rabies tracing experiments and with drop fix for electrolytic lesions retrieval from the drive-implanted mice. After being left overnight at 4 °C, brains were sectioned coronally at 100 µm thickness with a floating section vibratome (Leica VT1000s), washed in PBS and then labeled with 1:1000 DAPI solution (62248; Thermo Fisher Scientific). All sections were mounted and coverslipped with clear-mount with tris buffer (17985–12; Electron Microscopy Sciences). Confocal images were captured using a Leica TCS SP8 microscope with a 10 X objective (NA 0.40) and a Zeiss LSM 710 with a 10 x objective (NA 0.45). ML and DV coordinates for cortical tetrode rotations (*Figure 1—figure supplement 1*) were measured in ImageJ/FIJI (National Institutes of Health) from the midline and the pia to the center of the lesions and aligned to 4 matching coronal slices from the Mouse Brain Atlas (Allen Institute) for AP axis reference.

## Electrophysiology and data acquisition

Electrophysiology signals were acquired continuously at 30 kHz, while the behavioral tracking was acquired at 30 Hz with one or two lighthouse tracking stations (HTC Vive Base Station, Amazon). An additional camera (FFY-U3-16S2M-S, FLIR) was placed on the ceiling of the behavior box for behavior monitoring. Three mice were recorded on a first generation Open-Ephys system (*Siegle et al., 2017*) with an Intan 64 or Intan 32 (for the carbon fiber-implanted mouse) headstage. In these mice, tracking provided by two lighthouse receivers (TS4231, Digikey) attached at the base of the headstage, whose signal was recorded and powered through a teensy 3.6. The other nine mice were recorded on a new-generation Open-Ephys ONIX (*Newman et al., 2023*) system with 64 channel headstages with a powered commutator, that integrated electrophysiology and behavior tracking using the Bonsai software (*Lopes et al., 2015*).

Spikes were sorted on 300–6000 Hz band pass filtered continuous traces, using MountainSort (https://github.com/flatironinstitute/mountainsort, copy archived at *flatironinstitute, 2023*; *Chung et al., 2017*). Units were then manually selected based on the spike template shapes resembling action potentials with asymmetric waveforms and interspike interval (ISI) distribution centered away from the refractory period (*Figure 1—figure supplement 2A–C* for samples of HD units in ADn and RSC with tetrode waveforms, autocorrelation histograms and tuning curves in polar coordinates and *Figure 1—figure supplement 2D* for Isolation, SNR and mean Hz of all selected units from sessions sampling unique ensembles based on tetrode advancement tracking). After implant surgery, ADn-targeting tetrodes were lowered until HD-coding cells were identified based on their tuning obtained from brief recordings, and RSC targeting tetrodes were slowly lowered until well-isolated units appeared. When at least two HD cells in ADn were first detected, recordings of cue rotation or cue-on-off sessions were collected over a minimum of 2 weeks and up to 8 months. Tetrodes in ADn and RSC were regularly moved by ~20–150 µm increments, followed by recordings of short stable cue-on sessions to verify if the yield was improved. To avoid sampling of the same units for HD neurons quantifications and spike correlations for monosynaptic connections, units from selected sessions based on the movement of tetrodes and the number of units recorded and from only one session for the carbon fiber-implanted mouse and one for the large were included in these analyses. At the end of the experiments, electrolytic lesions were obtained by passing positive and negative current (20–25 µA) on each electrode contact for 5 s with a stimulus isolator (A365RC, WPI) while the animal was under isoflurane-induced anesthesia. After 30–60 min of recovery, the brains were extracted for histology.

## Data analysis
### HD unit selection

HD was quantified as the relative orientation of two or three infrared lighthouse receivers present on the integrated headstage, after their (x,y) coordinates were linearly interpolated to align to the same 50 Hz timestamps. For each session, HD tuning curves were quantified as the histogram of the spike trains over HD angles of 10 degree bins divided by the occupancy. For HD unit selection and information metrics for other spatial correlates, data from a stable cue-on period was used. Information was calculated in bits/spike as

$$I = \frac{\sum_i^N \lambda(x_i) \times log_2 \left( \frac{\lambda(x_i)}{\lambda} \right) \times p(x_i)}{\lambda}$$

following the methods of *Skaggs, 1993* where *x* is the binned HD (n=36 bins), p(x) is the occupancy, and $\lambda$ is the mean firing rate and $\lambda(x_i)$ is the firing rate for each angular bin. Cells in ADn and RSC were selected as HD-coding if the amount of directional information, the resultant and the concentration of the smoothed tuning curve were more than the 98th percentile of the shuffle distribution and if these criteria were met on two distinct stable, cue on periods with substantial HD angles occupancy. Shuffling of the spikes was obtained by shifting 500 times the spike trains by random amounts with respect to the HD from tracking. The peak number was obtained from MATLAB's *findpeaks.m*, with a minimum peak distance of 120°, width of 40° and prominence more than 11. For units with more than one peak (*Figure 1—figure supplement 3A*), we applied MATLAB's *fitnlm.m* with a basic von Mises model function with one peak

$$coeff_1 + coeff_2 \times \frac{1}{2pi * besseli(0, coeff_4)} \times e^{coeff_4} \times cos(angles - coeff_3)$$

where *besseli* is the modified Bessel function of the first kind, *angles* is the range of possible angles between 0 and *2pi* in 3600 bins, and the subscripted coefficients correspond to: 1 a baseline constant offset, 2 a scaling factor for the peak height, 3 the peak location, 4 the concentration parameter in the Von Mises probability distribution. For instances where up to 3 peaks were identified, the model function was expanded with a linear sum combining additional sets of height, location and concentration coefficients. Starting values for coefficients estimation were obtained from the *findpeaks.m* and 0 for the constant offset. The aim of this strategy was to identify a von Mises distribution anchored to the largest peak in tuning curves, whose resultant would have otherwise been much lower despite a strong directional information (*Figure 1E* 3rd example from top, and *Figure 1—figure supplement 3A*). Angular velocity was calculated as the first derivative in the unwrapped, down-sampled HD to 500ms and then re-interpolated in the 50 Hz temporal resolution.

## HD decoding

We decoded HD using a linear-Gaussian GLM based on the 20 ms-binned firing rates of ADn and RSC neurons, separately (*Figure 2—figure supplement 1A*). A Butterworth filter with cutoff normalized frequency of 0.2 was applied to the firing rates, which were then normalized. Maximum a posteriori estimation coefficients of the neuronal ensembles (the predictors) were obtained via ridge regression regularization was applied to the sine and cosine of HD, binned in 10° bins. The segments for the training were taken from a cue-on period at least 50 s away from rotation. Decoding was performed using MATLAB's *glmval.m* with the corresponding identity link function and HD reconstructed as the $tan^{-1}$ from the decoded bins. With this strategy, we obtained the HD representation, which was linked to the neuronal ensembles via the learned coefficients during training, in a test period of 120 s following the training session and in the period after cue rotation. For testing drift in dark and light, shorter training sessions during a stable cue-on (up to 8 min of data) were used and evaluated on the subsequent cue-on period and after the cue was turned off.

Results from HD decoding with the GLM were replicated with Long-Short-Term-Memory (LSTM) network composed of a single LSTM layer of 50 units and a fully connected layer into a cyclical MSE output with 36 possible HD values by wrapping around values to 2pi. The network was trained with MATLAB's deep learning toolbox with every new trial to predict the test and the realignment period. Because of the slightly better accuracy of the GLM in the test period, we chose to present the GLM results in the main figures.

## Detection of putative functional monosynaptic connections

We performed spike cross-correlation between all unique possible pairs of simultaneously recorded ADn-RSC neurons to detect putative monosynaptic connections. Excitatory connections appear as peaks in the cross-correlogram in the short time scale (1–5ms) above baseline (*Fujisawa et al., 2008*; *Stark and Abeles, 2009*). We focused on excitatory connectivity since corticothalamic and thalamo-cortical projections are excitatory. Cross correlograms were constructed in bins of 0.5ms by taking all

spikes occurring during cue-on trials in a session (or only during cue-off trials). The baseline correlation, simulating homogeneous firing, was constructed by convolving the cross correlogram with a 10 ms s.d. Gaussian window. Significant connections were detected if at least 2 consecutive bins in the 1–5ms window of the cross correlogram were above the 99.9th percentile of the cumulative Poisson distribution at the baseline rate.

## Interneuron and pyramidal neurons classification

Fast spiking (FS) interneurons and regular spiking (RS, pyramidal) neurons have distinct features that appear on the extracellular spike waveforms and can be used for classification (*Barthó et al., 2004*; *Wilson and McNaughton, 1993*). We applied the metrics described in *Sirota et al., 2008* on spike waveforms identified from the bandpass filtered continuous traces. Briefly, a mean spike waveform was obtained for each cortical neuron and the peak-to-trough was quantified as the time between the peak of the spike and the maximum point in the after hyperpolarization, whereas the symmetry around the spike was calculated as the difference between the height at the maximum point after spike peak and the maximum point before spike peak, divided by the sum of these two quantities. While in our dataset the symmetry value was unimodally distributed, the peak-to-trough was clearly bimodally distributed, allowing to cluster FS and RS with a previously reported (*Peyrache et al., 2015*) cutoff duration of 0.42ms, which resulted in average spike waveforms with a slow repolarization decay for RS and faster repolarization in FS (*Figure 4—figure supplement 2*).

## Quantification and statistical analysis

All statistical analyses were performed in MATLAB (MathWorks, R2020a). All spiking and behavioral data, with exception of the spike times for the detection of monosynaptic connections, was binned in 20ms bins. Behavioral tracking from the light receivers was linearly interpolated. Circular Statistic Toolbox (*Berens, 2009*) functions were employed such as circular means for quantifications of HD units' preferred firing directions (PFD), mean ensemble rotations and peak offsets, as well as neural population and decoded errors mean rotations and drifts, confidence intervals, tests of uniformity of angles, and circular correlations between decoded errors, PFD offsets of HD cell pairs between trials, PFDs within trials and mean ensemble rotations. Where the data was not circular, such as absolute values of angles, standard metrics and Pearson correlations coefficient were calculated.

Data in *Figure 1F–G* and *Figure 1—figure supplement 3G* included only 1 trial per selected sessions with unique ensemble cells, where significant tuning curves rotations were observed and finally where each direction bin had a 1 s minimum occupancy. This was to avoid smoothing PFDs offsets by averaging across before-after rotation segments. For comparisons of HD units' properties between cue-on and cue-off in *Figure 1—figure supplement 3H–L* and in *Figure 1—figure supplement 3F*, metrics for individual units were obtained by averaging across trials within the same session, which were again selected to sample unique ensembles following tetrode movements and changes in recorded unit numbers. A similar session selection criterion was used for the functional connectivity, where unique combinations of ADn-RSC ensembles were used. For *Figures 2D&E and 3D–G* and *Figure 2—figure supplement 1C–F*, *Figure 2—figure supplement 3A–G*, and *Figure 3—figure supplement 1A*&B the metrics of trials from largely overlapping ensembles were averaged and the statistics were performed on unique ensembles. In *Figures 1H and 2B* and *Figure 2—figure supplement 1G, H* data was presented as mean and standard errors of the rotation values from overlapping ensembles and between ranges of large positive, small positive, null, large negative and small negative and large negative rotations.

The difference in decoded HD or decoded HD errors in *Figure 2—figure supplement 3* was calculated as the median of the absolute medians of the wrapped difference in decoded HD (or HD error) of 7, non-overlapping 5 s segments, to cover a total of 35 s of stable test and realignment following cue rotation periods. Absolute medians from trials from the same ensembles were combined.

Two-tailed Kolmogorov Smirnov tests were used to compare the time lags of peak correlation between ADn and RSC decoded errors vs the null distribution obtained from 100 shuffles, and Wilcoxon Signed-Rank tests to compare the peak correlation values before and after cue rotation. Shuffle distributions for the decoded rotations in ADn and RSC (*Figure 2—figure supplement 1G*&H) were obtained by decoding the HD from circularly shifted spikes by random amounts 100 times. For analyses on decoded HD, trials were included if the decoding accuracy from that session was above

90% of that from shuffle spikes. Shuffle distributions for the decoded errors (both for dark and cue-rotation drifts) were obtained by circularly shifting the tracked HD by random amounts 100 times and subtracting it from the RSC decoded HD. Significant ensemble preferred directions rotations were determined if at least half of the HD cells experienced preferred direction shifts larger than the 98th percentile of a distribution obtained by randomly reassigning 500 times the indices around that rotation trial. p-Value thresholds of 0.05 were used for statistical non-parametric tests. Multiple comparison tests were performed with Bonferroni-correction.

## Acknowledgements

We thank Hongkui Zeng, Shenqin Yao, Ali Cetin, and the Allen Institute as well as Ian Wickersham and Heather Sullivan for sharing monosynaptic rabies tracing viral constructs. We thank Mila Halgren and Lukas Fischer for feedback on the manuscript and members of the Harnett laboratory and Wisam Reid for constructive criticism on the project. This work was supported by a MathWorks Graduate Fellowship (MSH v.d G), NIH K99 6943778 (JV), RO1NS106031 (MTH), the James W and Patricia T Poitras Fund at MIT (MTH), and the Klingenstein-Simons Fellowship Program (MTH).

## Additional information

### Competing interests

Jakob Voigts: The author is a board member of Open Ephys Inc, a public benefit workers cooperative in Atlanta GA. Jonathan P Newman: The author is president of Open Ephys Inc, a public benefit workers cooperative in Atlanta GA. The other authors declare that no competing interests exist.

### Funding

| Funder | Grant reference number | Author |
| --- | --- | --- |
| National Institutes of Health | RO1NS106031 | Mark T Harnett |
| Massachusetts Institute of Technology | James W. and Patricia T. Poitras Fund | Mark T Harnett |
| Esther A. and Joseph Klingenstein Fund | Simons Fellowship Program | Mark T Harnett |
| National Institutes of Health | K99 6943778 | Jakob Voigts |
| Massachusetts Institute of Technology | MathWorks Science Fellowship | Marie-Sophie H van der Goes |

The funders had no role in study design, data collection and interpretation, or the decision to submit the work for publication.

### Author contributions

Marie-Sophie H van der Goes, Conceptualization, Resources, Data curation, Software, Formal analysis, Validation, Investigation, Visualization, Methodology, Writing - original draft, Writing – review and editing; Jakob Voigts, Jonathan P Newman, Resources, Software, Methodology, Writing – review and editing; Enrique HS Toloza, Formal analysis, Writing – review and editing; Norma J Brown, Investigation; Pranav Murugan, Formal analysis; Mark T Harnett, Conceptualization, Supervision, Funding acquisition, Project administration, Writing – review and editing

### Author ORCIDs

Marie-Sophie H van der Goes  http://orcid.org/0000-0001-5251-6664
Jonathan P Newman  https://orcid.org/0000-0002-5425-3340
Mark T Harnett  https://orcid.org/0000-0002-5301-1139

### Ethics

All animal procedures were performed in accordance with NIH and MassachusettsInstitute of Technology Committee on Animal care guidelines (protocol number 0521-036-24).

### Decision letter and Author response

Decision letter https://doi.org/10.7554/eLife.82952.sa1
Author response https://doi.org/10.7554/eLife.82952.sa2

---

## Additional files

### Supplementary files

• MDAR checklist

• Supplementary file 1. Recording Sessions Summary.

### Data availability

Data and associated code have been uploaded on Dryad at https://doi.org/10.5061/dryad.dfn2z3555.

The following dataset was generated:

| Author(s) | Year | Dataset title | Dataset URL | Database and Identifier |
|---|---|---|---|---|
| van der Goes M, Voigts J, Newman J, Toloza E, Brown N, Murugan P, Harnett MT | 2023 | Coordinated Head Direction Representations in Mouse Anterodorsal Thalamic Nucleus and Retrosplenial Cortex | https://doi.org/10.5061/dryad.dfn2z3555 | Dryad Digital Repository, 10.5061/dryad.dfn2z3555 |

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
