## [Editor Report]

This useful study investigates the coordination of neurons coding for head direction in the anterior thalamus and the retrosplenial cortex during environmental manipulations. The evidence supporting the claims of the authors is solid. The paper will be of interest to neuroscientists working on spatial navigation.

---

## [Decision Letter]

**Decision letter after peer review:**

Thank you for submitting your article "Coordinated Head Direction Representations in Mouse Anterodorsal Thalamic Nucleus and Retrosplenial Cortex" for consideration by *eLife*. Your article has been reviewed by 3 peer reviewers, and the evaluation has been overseen by a Reviewing Editor and Laura Colgin as the Senior Editor. The reviewers have opted to remain anonymous.

Essential revisions:

While each reviewer has raised a number of specific concerns about the present study, there was an agreement that the following essential revisions needed to be addressed to warrant the publication of the manuscript.

1) The study claims that the 0-lag correlation in decoding error demonstrates a near-synchronous encoding of HD in the AD-RSC network. However, this observation may arise from erroneous decoding and video tracking. Specifically, the decoding seems quite unreliable at times, and concomitant errors in HD tracking and/or decoding would lead to a high 0-lag correlation. This problem can be addressed by measuring correlation at times of reliable decoding only and possibly using decoding techniques depending less on the neuron's tuning curves (i.e. unsupervised techniques).

2) Cue rotation did not necessarily lead to a shift in the internal HD signal. It seems like the animals after some time ignored the position changes of the visual cue. In this context, it doesn't make much sense to qualify the update of the HD reference as "successful". It would perhaps be better to include the first rotations or include only the trials that led to a substantial rotation of the internal HD.

3) There are some concerns about the quality of the spike sorting and criteria used for the identification of HD cells, as example tuning curves seem broader than what has been previously reported. Additional examples and quantification of sorting quality could address this problem.

4) Considering how critical the number of simultaneously recorded neurons is to evaluate the reliability of decoding, the study should include a detailed table of the number of neurons per session, etc.

5) Along the same lines, whether recordings at the same tetrode depth were included as independent samples is unclear. A detailed table of sessions, tetrode position, number of cells, etc. would be very informative.

6) The limitations regarding RSC-to-AD connectivity should be carefully discussed as RSC projecting neurons are likely in layer 6 only and may not have been sampled as well as other layers. Furthermore, only one tracing technique was used, not ruling out the possibility that these viruses have a poor tropism for this specific pathway.

The authors are invited to address as much as possible specific comments from the reviewers, appended below.

*Reviewer #1 (Recommendations for the authors):*

While I believe there are very strong points to this manuscript as noted above, some concerns about the experimental design have dampened my enthusiasm a bit. I have a series of specific questions.

1. The design of the behavioral protocols is not ideal to draw conclusions on visual landmark updating, as the visual cue was not presented as a reliable landmark for the animal. (It had shifted repeatedly while the animal was in the arena, and therefore probably underwent 'devaluation' (line 428) over time, and 90{degree sign} cue rotations often gave rotations of the neural representation of HD around 0 (cf line 1069)). If I understand correctly, this means that the animals after some time ignored the position changes of the visual cue. In this context, it doesn't make much sense to qualify the update of the HD reference as "successful" (line 261), after cue rotation, when an HD shift of > x{degree sign} is observed. It is a mismatch situation, and the mouse might rely more on self-information, proprioceptive and vestibular, than on the moving visual cue. I suggest changing the wording ("successful"). Beyond that, could it be helpful to include only the first few cue rotation experiments for a given animal? Before it considers the visual cue as unreliable.

There might be an opportunity to investigate more deeply the effects of learning cue reliability (or unreliability) over time.

2. Can you rule out potentially confounding effects of recordings obtained during early or late phases of the repeated exposures to the visual stimuli, that would affect the coherence between the AD and RSC HD representations? Are there more or less HD cells over time, when the animals have more prior experience with the environment?

3. To help the reader to better understand how the experiments were structured, I suggest including an overview table, indicating, per mouse, for each of the 12 mice of this study:

Mouse id;

Tetrode positions in Suppl Figure 1;

Which experimental protocols were run;

How many sessions/trials;

How many units in AD and/or in RSC were recorded;

How many of them were HD or nonHD cells;

Number of possible AD – AD pairs;

Number of possible RSC – RSC pairs;

Number of possible AD – RSC pairs.

4. Often times persistence of directional firing in the dark is used as a criterion for Head direction firing, and it may be useful to distinguish HD cells from visually responsive cells for example. I saw that 3 mice were tested in the dark (Figure 3), does it mean 9 mice were not? (How sure would you be to qualify directionally tuned cells as HD cells, and not visually responsive cells, if a dark condition was missing?)

Furthermore, the criteria for HD cells typically include cue control. Because of the lack of a disorientation period during the cue rotation, this may be difficult to affirm for the protocols used here. Do you think it is still justified to qualify all the directionally tuned neurons as HD cells?

Did you examine if units identified as HD might change and become nonHD (or vice versa) across different recording conditions?

In the dark recordings, do you find that some directionally tuned cells become silent, and might those be cells responding to the visual cue?

5. Please show more examples of tuning curves of HD and nonHD cells in addition to those in Figure 1B. I would find it more convincing and also a helpful resource, for reference, to see the range of different shapes of the tuning curves in AD and in RSC, their width, and peak firing rates.

How many HD cells have more than 1 peak (suppl Figure 2A), in AD and in RSC?

6. All head direction cells were pooled together to produce suppl Figure 2D. In an ideal setting, one would expect the diagonal of preferred peak firing directions to be straight and to show uniform coverage of the 360{degree sign}, at least for ADn. This is not entirely the case. Could some of the HD cells be cells that are tuned to the visual cue? Can you indicate the cue position(s) on the graph, and might they be overrepresented?

7. Decoding: how many neurons were included? A range is given in the Discussion section, line 423, this information should rather be moved to the methods, and the actual number of simultaneously recorded units for each ensemble (Figures2, 3, S6) indicated in the results (or figure legends). How may this number influence the accuracy of the decoding?

8. What is the firing frequency in light and in dark (Figure 3, suppl Figure 6), and is decoding still reliably carried out at low firing frequencies?

9. Figure 2C. It may be misleading to label the Y axis as a 'decoded error' with respect to a visual cue when the visual cue might not function as a visual landmark (see point 1, the mouse might rely more on self-information, proprioceptive and vestibular, than the visual cue). Figure 2C would benefit from showing more time before 0 (baseline).

10. Line 359 states that AD-to-RSC connectivity was divergent, but in Figure 4H it appears that four differently tuned AD neurons, in blue, contact two (same?) RSC neurons (same-shaped tuning curves 1-2, and 3-4). Is this a mistake? Also, can you confirm that the RSC (red) traces all qualified as HD units (it looks quite untuned by the eye)?

11. In addition to the connectivity rates indicated in Figure 4, I would suggest also including the numbers for ADn-ADn and RSC-RSC pairwise connectivity rates, for comparison, and for the sake of completeness.

12. line 274, "highly synchronized" – please indicate the time scale.

13. It is not easy to read the color codes in some instances, including Figure 4G FS vs RS; especially when histograms overlap, also in Figure S2BC and Figure S6A.

14. Figure 4 rabies tracing: There is only one example for each, please indicate n = 1, unless there are more experiments that are not shown (in which case, please show them in the supplementary data). In B, why are the presynaptically labelled cells in RSC red? Shouldn't they be green?

Figure 4G: Is there really similar tuning between connected Ad and RSC HD units (line 451)? Even if the circular mean of the RS peak differences is at -7.44{degree sign} (Figure 4G), very few pairs seem to have such a small difference in preferred head direction.

15. Suppl F1b: Why are there so many tetrode locations in RSC, the text said 11 mice were recorded in RSC. Give AP levels for thalamic sites, to help the reader to situate the lesions.

16. Mouse numbers:

Line 74, 9 mice were simultaneously recorded;

Line 97, 8 mice… which is correct?

*Reviewer #2 (Recommendations for the authors):*

1. The authors state that 'our results provide direct evidence against the hypothesis that visually guided updates in the HD reference would first appear in the RSC' (page 17 lines 416ff). This is indeed a strong statement, but I am not sure it is entirely supported by the authors' data. I wonder if a temporal offset in cue-rotation responses between AD and RSC HD cells might have gone undetected in the authors' study (see also point 2 below). For example, the visual update drive – in the form of e.g. an excitatory input from RCS that realigns HD cells in the AD- could be difficult to detect, considering the temporal resolution of the authors' experimental design. Hence, it is difficult to prove a 'negative result' here. The authors refer to two possible limitations (page 20 lines 491-492): 'cue devaluation with repeated trials' and 'temporal control of the cue rotation'. Is it conceivable that these limitations might have prevented the observation of a temporal delay between the shifts in HD representations between retrosplenial and AD?

2. It is surprising that the authors recorded in the retrosplenial cortex, but do not comment upon nor describe bidirectional HD cells (Jakob et al., 2017). Since these neurons are regarded as prime candidates for the sensory-HD integration the authors aim to study, I think they should focus their analysis on these firing patterns specifically.

3. The tuning curves of HD cells in the AD nucleus are very broad (e.g. Figure S2E, Figure 1B, E), which seems to be at odds with prior literature, and this is potentially concerning. I wonder if this might be accounted for by the authors' definition of HD cells, and/or by spike sorting quality. The authors should try to quantify and exclude systematic spike sorting issues. As a 'control' they should maybe apply more stringent inclusion criteria, restrict the analysis to 'good' sharply tuned and nicely sorted HD units, and see if their conclusions hold for this refined dataset as well. They should also comment on whether and how (possibly suboptimal?) sorting might have impacted the 'negative result' of the present study, i.e. the fact that against predictions from prior literature, the authors did not find evidence for a visual update drive in the RSC cortex.

4. The authors state that "the RSC is not wired to drive visual reference update" (page 17 line 413). However, connectivity RSC◊ AD has been demonstrated in several prior studies, and this is somewhat not recapitulated by the authors' tracing experiments with rabies viruses (I wonder if this finding needs to be confirmed with more conventional tracers, to exclude suboptimal tropism or synapse-specific effects of the viral transynaptic tracing, e.g. Rogers and Beier J Neurosc Methods 2021). The authors' statement is also supported by functional connectivity data (cross-correlations). The 'apparent sparsity' of connectivity could however be biased by cellular sampling since most thalamic projecting neurons are expected to be found in the deep layers (mostly L6) which seem not to have been systematically sampled with tetrode recordings.

5. The authors state that they have performed 'widespread sampling of RCS locations' (page 18 line 449). This is to some extent true; however, the retrosplenial cortex goes well beyond the recording locations sampled by the authors. I think the authors should reword these statements and acknowledge the possibility that other subfields of the retrosplenial cortex (which were not extensively sampled by the authors) could in principle be responsible for the visual update drive to the thalamus.

*Reviewer #3 (Recommendations for the authors):*

Decoding errors: Average decoding error in RSC seems very high – 90.83 degrees for fast head turns seems like a chance level and I am not sure if it is still appropriate to talk about successful decoding of HD in that case. Since the quality of decoding is critically dependent on the number and tuning of HD cells recorded in each session, could the authors provide evidence that in each session included in the dataset both ADN and RSC decoders perform significantly better than chance as estimated from spike shuffles?

Examples in Figure 2A show many gaps in the tracked HD of the mouse, which to me indicates the sub-optimal quality of the behavioural tracking. This is especially important for analyses of decoding errors like the one in Figure 2D that shows that internal HD representations in ADN and RSC are coordinated at zero lag (+/- 20ms). The observed zero-lag peak could be instead explained by errors in behavioural tracking dominating the analysis, which would affect both representations simultaneously and show spurious zero-lag positive correlations. However, coordination could also be shown by computing the difference between internal HD decoded from ADN and RSC at different time lags, without reference to the HD tracked behaviourally. Could the authors include this sanity check in the manuscript?

The percentage of HD cells in RSC reported in the literature is generally low (~10%), but those RSC HD cells often show very narrow HD tuning. Yet, judging by the examples, the HD tuning of RSC cells recorded in the study seems worse than expected. Could the authors provide some statistical measure of HD tuning stability for each cell (i.e. correlating the first and second half of the baseline recording) as a sanity check? Canonical HD cells should show very good tuning stability under constant conditions.

Supplementary Figure 2F seems to show an overrepresentation of HD cells with similar preferred directions, especially so in RSC. If this is indeed the case, I am wondering if this apparent HD tuning could be explained by any other behavioural variable that in this session happens to be correlated with HD, as in my experience sometimes happens when a tuning curve shows weak HD tuning. I find that such spurious HD tuning often disappears if only epochs when the animal is moving are included in computing the HD curve, could the authors demonstrate that HD tuning does not change if stationary epochs are excluded?

Could the authors provide the breakdown of overall cell numbers recorded per animal (including nonHD/HD cells), for example as a table?

Are there any differences in HD modulation across layers and sub-regions of RSC?

Figure 1J shows that HD cell receptive fields exhibit both clockwise and counterclockwise rotations, do these correspond to the clockwise or anticlockwise rotations of the cue, or is the realignment not dependent on the direction of cue rotation?

The manuscript often uses a number of trials as their sample size for statistical analyses and the methods state that tetrodes were regularly advanced, but there is no indication of whether multiple trials at the same tetrode position were included in the same statistical comparison (except for recordings '4 days apart' for the HD tuning and synaptic connectivity analyses). Multiple trials with a high likelihood of recording the same cell population should not be counted as separate samples when calculating statistical significance. The authors should clarify whether tetrodes were moved between recording sessions and, if that was not the case, correct their statistical analyses to, e.g. perform statistical tests on average values per recorded population over multiple sessions.

[Editors' note: further revisions were suggested prior to acceptance, as described below.]

Thank you for resubmitting your work entitled "Coordinated Head Direction Representations in Mouse Anterodorsal Thalamic Nucleus and Retrosplenial Cortex" for further consideration by *eLife*. Your revised article has been evaluated by Laura Colgin (Senior Editor) and a Reviewing Editor.

The manuscript has been improved but reviewer #2 has some remaining issues that need to be addressed (mostly clarification of result interpretation and discussion), as outlined below.

*Reviewer #2 (Recommendations for the authors):*

In response to my concern #1, the authors state: "We agree that the 20ms temporal resolution prevents the observation of a temporal offset occurring at shorter timescales, for example, that of direct excitatory drive between the two regions". This potential limitation should be specifically acknowledged in the discussion, e.g. by stating that the temporal resolution of the decoding approach (20 ms) might have prevented the resolution of faster (monosynaptic) dynamics, which are approx one order of magnitude faster. Again, I believe that saying "our results provide direct evidence against the hypothesis that visually guided updates in the HD reference would first appear in the RSC" (lines 343ff) is a very strong statement. The authors provide evidence consistent with this hypothesis but do not formally rule out alternative updating mechanisms that might occur on faster timescales, i.e. beyond the authors' temporal resolution. Faster dynamics (<20ms) might have gone undetected in the authors' work. This limitation should at least be acknowledged in the discussion.

The sentence in the abstract "with surprisingly little reciprocal drive in the corticothalamic direction" is not supported by the authors' data, nor by previous anatomical work showing strong RS deep layer-to-ADn connectivity. This is also clear from the authors' tracing data (from retrograde tracings, RS L6 shows up prominently). At some points, the authors make granular vs agranular RS distinctions (e.g. lines 377ff), but this is not done consistently throughout the manuscript, and the above sentence in the abstract generally refers to RS. The authors also refer to "asymmetry" in RS – ADn connectivity; I do not see what the authors mean by 'asymmetry', since the connectivity follows the classical thalamocortical rules (e.g. L6 back to thalamus). As for the functional data, more extensive sampling of RS L6 is needed to support claims about asymmetric connectivity. Hence, such strong claims (as the one in the abstract) should at least be moderated throughout the manuscript.

---

## [Author Response]

Essential revisions:While each reviewer has raised a number of specific concerns about the present study, there was an agreement that the following essential revisions needed to be addressed to warrant the publication of the manuscript.1) The study claims that the 0-lag correlation in decoding error demonstrates a near-synchronous encoding of HD in the AD-RSC network. However, this observation may arise from erroneous decoding and video tracking. Specifically, the decoding seems quite unreliable at times, and concomitant errors in HD tracking and/or decoding would lead to a high 0-lag correlation. This problem can be addressed by measuring correlation at times of reliable decoding only and possibly using decoding techniques depending less on the neuron's tuning curves (i.e. unsupervised techniques).

We have addressed this concern in multiple ways:

1) We first changed the overall metric for decoding accuracy to the median of the absolute decoded error instead of the 75^th^ percentile. This new metric, which better reflects the reader’s expectation, is much lower than what we previously showed, and more comparable to other studies.

2) For our cross-correlation analysis, we have corroborated the coordination results with a revised rotation threshold and number of independent ensembles in Figure 2D and E. We also performed the same analysis by excluding points of low (below 30°/second, Author response image 1 and B) or high (above 30°/second, Author response image 1 and D) angular velocity (AV). Higher AV is associated with a small but significant reduction in decoding accuracy (shown in Figure 2 – Supplement 1E and F), thus suggesting a more reliable quantification of the lag between the two regions when considering only time points with low AV. However, the exclusion of several data points, on average halved at 0-time lags and further exacerbated at lags away from zero, resulted in higher variability in the cross-correlation traces (Author response image 1). This variability was more pronounced when we considered points of high AV (Author response image 1). In both cases, the average correlation traces pointed to a peak at 0-time lags, more pronounced in the high AV case and much smoother in the low AV case, as expected, and the histograms of the peaks showed a majority at 0-time lag.

**Author response image 1. sa2fig1:** Correlation of decoded HD errors at different angular velocity. (**A**): Individual trials (grey traces, n=108) and averages of the unique ensembles (n=28) of the cross correlations between decoded errors composed only of the time points with high AV (more than 30°/s). (**B**): Distribution of the time lags corresponding to the cross-correlation peaks from A, in 20 ms bins. Wilcoxon Signed-Rank test between before and after rotation p=0.537 and two-sample Kolmogorov-Smirnov test, p<0.0001 for both stable and shifted versus the null distributions in grey, n=28 ensembles. (**C**): Same as A, but for decoded errors composed of time points with low AV. (**D**): Same as **B**, but for the peaks in **C**. Wilcoxon Signed-Rank test between before and after rotation p=0.131 and two-sample Kolmogorov-Smirnov test, p<0.05 for both stable and shifted versus the null distributions in grey, n=28 ensembles.

3) Reviewer 3 suggested an alternative method to the cross correlation in order to test for the coordination between the two regions, namely comparing the difference between the decoded HD of the two regions to that at different time lags. We did this and present the results in Figure 2 – Supplement 3B. Each trial (n=108, grey traces) is the median of the median absolute difference between ADn and RSC decoded HD calculated for sliding 5s windows of 35s decoded traces at up to +/- 5s time lags. This approach was taken to circumvent the possible problems of leading or lagging of the decoded HD depending on the head rotations (clockwise or counterclockwise) and the overall smearing of possible temporal offsets for long traces. However, in contrast to the cross correlation of the decoded error, the difference on the decoded HD does not consider opposing scenarios of lead/lag of the HD representations following positive and negative cue rotations. Therefore, we still resorted to the difference of the decoded HD errors (with some flipped to be on the same sign of the rotations), in Figure 2 – Supplement 3C-F. As in Figure 2 – Supplement 3B and as a mirror image of the cross-correlation results, we observed a trough, in other words a point of minimum error between the two regions, at 0-time lags, both on the individual (Figure 2 – Supplement 3C and D) as well as on the unique ensembles (Figure 2 – Supplement 3E and F) traces averages, resulting in a majority of 0-time lags of the trough histograms both during stable cues and after cue rotation.

4) We have replicated the cross-correlation results by using a different decoding strategy, namely a long-short-term memory network (LSTM) with a single hidden layer of 50 units trained on the neural activity of each trial, which, as suggested in the main points of the reviewers, relies less on the tuning curves of the individual units. We included this approach in the method section of the manuscript but decided to present the GLM decoding results in the main manuscript because of the overall higher decoding accuracy in the test period observed with the GLM. In Author response image 2 similar panels as those shown throughout the manuscript for this decoding method, with a detailed legend. We note that at times the number of ensembles or trials is different from that of the GLM-results because the spike shuffling control was not applied in this instance and because averages based on the value of the decoded rotations might differ from those obtained with the GLM-decoder.

In the course of the revisions we have also implemented several other changes that have improved the quality of the decoding and the robustness of the results: (a) We have extended the test period used for quantifying this accuracy from 80s to 120 s; (b) For our GLM decoding, we have excluded sessions that did not pass our threshold for decoding accuracy in comparison to spike shuffling for individual sessions, as suggested by reviewer 3; (c) We have revisited the dataset with stricter criteria for units selection and inclusion of sessions and trials that had good heading occupancy; (d) We have averaged across trials from the same sessions and from sessions with overlapping recorded ensembles to yield statistical comparisons between independent samples.

Altogether, our new analyses and substantial refinements to both the data inclusion criteria and our existing analyses addresses the concerns about spurious correlations, and demonstrates the robustness of our results.

**Author response image 2. sa2fig2:** LSTM-based decoding of HD recapitulates GLM-based decoding results. (**A**): Mean and 95% CI of the absolute decoding error distribution for ADn (blue) and RSC (red) for each region from the test period from rotation trials (n=37 unique ensembles ADn, N=30 for RSC). (**B**): Violin plots of the medians of the absolute decoded errors for ADn and RSC ensembles (p<0.001, Mann-Whitney test, medians highlighted in black 35.69° in ADn, n=37, and 50.88° in RSC, n=30). (**C**): Left: scatter plot with error bars of the mean rotations from simultaneous ADn HD neurons tuning curves versus the mean rotations calculated from decoding HD from ADn neurons (circular correlation coefficient=0.82, p<0.0001, n=114 ensembles averaged across trials of positive and negative small and large rotations). Right: Same as **E** but for RSC (circular correlation coefficient=0.80, p<0.0001, n=60 ensembles). (**D**): (left) Correlation of the decoded rotations from simultaneous ADn and RSC ensembles (circular correlation coefficient r=0.58, p<0.001, n=97 trials averaged across the same ensembles and for similar small positive, small negative, large positive, large negative and null rotations, from 8 mice).(Right) the correlation value is above the 99^th^ of 100 correlation values calculated for shuffled RSC rotation values. (**E**): Cross correlations of ADn vs RSC decoded HD errors are on average highest at 0 s time lags both in the 75s stable test period (left) and in the 75s after cue rotation (grey traces are the individual trials, n=110, colored black and green are the mean and the 95% CI of the unique ensembles, n=31). (**F**): Distributions of time lags corresponding to the cross-correlation peaks from the ensemble traces from stable periods (left) and after cue rotation (right) vs in grey the distributions from cross correlations with 100 times shuffled RSC decoded errors (n=31, p<0.0001 Kolmogorov-Smirnov test between the true and the shuffled distributions for both stable and realignment data, p=0.832 Wilcoxon Signed-Rank test between stable and realignment time lags of peak correlations). Insets show the same distributions zoomed in the -0.5 to 0.5 s range. Left y-axis and right y-axis are on different scales to show the values of the correlations from shuffled trials. (**G**): Distributions of the time lags corresponding to the minimum absolute decoded error of the median differences shown in K (p=0.618 Wilcoxon Signed-Rank test between stable and realignment time lags of peak correlations). (**H**): Left, comparison of the correlation values for the 13 unique ensembles between cue on and cue off periods (Wilcoxon Signed-Rank test p=0.7353). Middle and right, the mean correlation values for cue on and cue off are above the 99^th^ of 100 mean correlations with shuffled RSC drifts.

2) Cue rotation did not necessarily lead to a shift in the internal HD signal. It seems like the animals after some time ignored the position changes of the visual cue. In this context, it doesn't make much sense to qualify the update of the HD reference as "successful". It would perhaps be better to include the first rotations or include only the trials that led to a substantial rotation of the internal HD.

We have eliminated from the text the word successful and changed it to “significant change”. We have also raised the threshold for detecting small rotations to 17.2°. To determine whether initial sessions/experiments are indeed more predictive of better cue control, we plotted for each mouse and for the two regions the neural rotations calculated from the difference between the decoded HD with the GLM and the tracked HD over the session numbers and we labeled the data points according to the size of the cue rotation (see response to specific point 1 of reviewer 1). While for some mice the size of the decoded rotation was reduced over time for others this trend was not obvious. We show in Author response image 3 the distributions of decoded rotations from ADn (for ADn or ADn-RSC simultaneous recordings) or RSC (for RSC only recordings) ensembles (left 2 graphs) and or the mean rotations from HD units (right 2 graphs) normalized by the cue rotation value (purple for small cue rotations, green for large cue rotations). Only rotations above an absolute minimum of 17.2° were included and in addition, for the decoded ensembles, only those that surpassed the accuracy threshold from spike shuffling. We can observe, similarly to the analysis presented in Figure 2 – Supplement 2B, that cue control is stronger for small cue rotations, as the rotations are clustered around 1, whereas the large cue rotations result in more variable, but nonetheless significant, HD realignment.

Screening the rotations combined with stricter quality parameters supports our subsequent analyses of the coordination between ADn and RSC.

**Author response image 3. sa2fig3:** Normalized decoded rotations. Left 2 graphs, distribution of the decoded rotations, normalized by the size of the cue rotation, from ADn ensembles, right 2 graphs from RSC ensembles. Purple indicates the small cue rotations, green the large cue rotations.

3) There are some concerns about the quality of the spike sorting and criteria used for the identification of HD cells, as example tuning curves seem broader than what has been previously reported. Additional examples and quantification of sorting quality could address this problem.

Units were manually selected based on the spike template shapes resembling action potentials with asymmetric waveform and afterhyperpolarization and inter-spike interval (ISI) distribution away from the refractory period. We have revisited the sorted units in our recordings and the HD selection criterion to be more stringent together with a more accurate sampling of non-overlapping recorded ensembles. As a result, the percentage of RSC HD units is lower than in the previous version of the manuscript, while that of ADn slightly increased, with medians for RSC HD group improved for directional information and resultant length (Figure 1 – Supplement 3B and C). In response to reviewers’ suggestions, we show several new analyses that confirm different properties of HD units in ADn and RSC. Specifically, in Figure 1 – Supplement 2F we show the stability of the HD peaks between two segments of a stable cue period and the similarity of the resultant between cue on and cue off and between cue rotations.

We have added more examples of tuning curves of ADn and RSC from different mice in Figure 1 – Supplement 2A-C, where we show the tuning curves of the same unit with the cue at two different angular positions. For each unit we also show the waveforms on the 4 channels of the tetrodes as means and standard errors and the spike auto correlograms in the -15 to 15 ms. In Figure 1 – Supplement 2C, we show different ADn units from the same session. To answer the concerns about spike sorting quality, we show in Figure 1 – Supplement 2D the distributions of noise overlap, isolation metrics and mean firing rates as extracted for each unit cluster from the mountainsort spike sorting. We labeled HD and Non-HD units from RSC and ADn and show that for these classes the distributions of the different metrics are largely overlapping.

4) Considering how critical the number of simultaneously recorded neurons is to evaluate the reliability of decoding, the study should include a detailed table of the number of neurons per session, etc.

We thank the reviewers for the suggestion and have added a table (Supplementary File 1) in which, for each mouse, we list the recording sessions and the session types, the tetrode locations and movements, the total number of units and the number of classified HD units in each region, the total number of possible ADn-RSC pairs and the number of connections. We also show in response to reviewer 1 the relationship between the number of recorded units and the quality of the decoding.

5) Along the same lines, whether recordings at the same tetrode depth were included as independent samples is unclear. A detailed table of sessions, tetrode position, number of cells, etc. would be very informative.

We have noted in Supplementary file 1 for each tetrode the fraction of turns done after the recording session. We have adjusted the criterion for identifying independent sessions to more closely follow these tetrode movements and the number of recorded units, which may change also as a result of drift and movement due to attaching/detaching of the headstage between recording sessions. This was done independently for RSC and ADn tetrodes. For Figure 1C,F,G,H, and Figure 1 – Supplement 3 and Figure 2 – Supplement 1, where HD unit quantifications, HD coding properties, decoding, and rotation accuracies are shown, the presented data comes from sessions sampling different units in each region according to the tetrode movement and the number of units recorded. For analyses addressing the regional coordination (in Figure 2 and Figure 2 – Supplement 3, Figure 3D,E,G, and Figure 3 – Supplement 1A and B) and the functional connectivity (Figure 4 and Figure 4 – Supplement 2), sessions where either RSC or ADn tetrodes where moved or the number of units substantially differed were counted as independent samples. In the cases of Figure 1H, 2B, Figure 2 – Supplement 1 F&G and in the above figure of the LSTM results panel E, averages of trials across the same ensembles were calculated by separating the individual the ADn rotation values according to whether they were large negative small negative, null, small positive and large positive.

6) The limitations regarding RSC-to-AD connectivity should be carefully discussed as RSC projecting neurons are likely in layer 6 only and may not have been sampled as well as other layers. Furthermore, only one tracing technique was used, not ruling out the possibility that these viruses have a poor tropism for this specific pathway.

We have extended the discussion on the RSC-to-ADn anatomical tracing data to consider potential tropism of the rabies virus and compare our results to past retrograde tracings using other techniques. Along those lines, we show in Figure 4 – Supplement 1B that injected retrobeads in ADn do label deep layers, particularly layer 6, of mostly granular RSC. The difference in density of the presynaptic labeling may result not only from viral tropism but also from the density of ADn labeling: in fact, the virus cocktail used for ADn rabies tracing consists of AAV1-Syn-Cre and two helper viruses, thus diluting the chances of co-infection necessary for retrograde labeling.

We acknowledge that a denser sampling of the deeper layers of granular RSC may provide a more definitive answer to the degree of functional connectivity between the two regions. We want to point out however that 9 out of 48 RSC tetrodes in mice with simultaneous ADn-RSC recordings were located in the deepest layers, and nonetheless only 4 connections were found. It is possible that we have missed more posterior hotspots in the granular RSC bordering postsubiculum, another region with known direct connections to ADn and with a functional role in controlling visual anchoring of ADn HD (Goodridge and Taube 1997). Regardless of the exact connectivity rate, a striking asymmetry exists between corticothalamic and thalamocortical connections, as previously assessed in postsubiculum (Peyrache et al. 2015). Moreover, our conclusions highlight that the most economical explanation to the coordination of the HD representations decoded from the ensemble activity we recorded relies on a feedforward drive of HD in the thalamocortical direction.

The authors are invited to address as much as possible specific comments from the reviewers, appended below.Reviewer #1 (Recommendations for the authors):While I believe there are very strong points to this manuscript as noted above, some concerns about the experimental design have dampened my enthusiasm a bit. I have a series of specific questions.1. The design of the behavioral protocols is not ideal to draw conclusions on visual landmark updating, as the visual cue was not presented as a reliable landmark for the animal. (It had shifted repeatedly while the animal was in the arena, and therefore probably underwent 'devaluation' (line 428) over time, and 90{degree sign} cue rotations often gave rotations of the neural representation of HD around 0 (cf line 1069)). If I understand correctly, this means that the animals after some time ignored the position changes of the visual cue. In this context, it doesn't make much sense to qualify the update of the HD reference as "successful" (line 261), after cue rotation, when an HD shift of > x{degree sign} is observed. It is a mismatch situation, and the mouse might rely more on self-information, proprioceptive and vestibular, than on the moving visual cue. I suggest changing the wording ("successful"). Beyond that, could it be helpful to include only the first few cue rotation experiments for a given animal? Before it considers the visual cue as unreliable.

We understand the reviewer’s concern about the behavioral design, however we also point out that other studies have approached the visual updating without disorientation before cue rotation and observed rotations (Knight et al. 2014 and, though the rotation was done on the animal through a rotating platform, also Knierim et al. 1998). Our behavioral design choice was also motivated by the possibility to closely compare the influence of the cue position on the changes of the neural activity, without adding other behavioral conditions. However, in light of the reduced cue control, especially for the 90° rotations, we have deleted the word “successful” update of the reference frame to “change”. In Author response image 4 we show for each mouse and for the two regions (ADn in panel A and RSC in panel B) the neural rotations calculated from the difference between the decoded and the tracked HD over the session numbers, While for some mice the size of the decoded rotation was reduced over time (see Mouse 4 both ADn, panel A, and RSC, panel B), as expected from a progressive devaluation of the cue, for others this trend was not obvious, suggesting that taking only the first rotations might not necessarily imply perfect HD rotations. We believe that this screening focused on the rotations, together with stricter quality parameters, can support the following analyses of the coordination between ADn and RSC.

**Author response image 4. sa2fig4:** Rotations over time. (**A**): Decoded rotations from ADn ensembles plotted over trials for each mouse. We have labeled the data points according to the size of the cue rotation (green for large cue rotations and purple for small cue rotations). The filled markers are trials included in the analyses presented in Figure 2 and in Figure 2 – Supplements 1, 2 and 3; open markers are trials rejected by the threshold for decoding accuracy in comparison with the decoded HD from shuffled spikes or by a decoded rotation less than 17.2°. (**B**): Same as **A**, but for RSC ensembles.

There might be an opportunity to investigate more deeply the effects of learning cue reliability (or unreliability) over time.

We thank the reviewer for raising this interesting point. Many previous studies have tested (Yoganarasimha and Knierim, 2006, Zugaro et al., 2003, Jeffery 1998, Knierim et al. 1998) or modeled (Yan, Burgess and Bicanski, 2021, Page and Jeffery, 2018) the conditions and the under which cues can be used to anchor the HD depending on their reliability. Sit and Goard, 2023 and previously Jacob et al. 2017 reported in RSC firing patterns and proposed possible circuit mechanisms that could process landmarks, however further experimental evidence of the conditions of reliability and processing of visual inputs are needed. We show that ensemble ADn and RSC cells rotate with the cue rotations, especially to the smaller cue rotations, where the mismatch with the internal HD is smaller (Figure 2 – Supplement 2B), but with large cue rotations, where the mismatch is larger, the HD reference does not always follow the cue rotations even though a realignment from the previous reference is observed, somewhat similarly to what Knight et al. 2014 observed. Besides being a topic that would require a separate research article, we believe that we don’t have the right dataset to fully explore the question of cue reliability for several reasons: (1) because we have presented only one visual cue in our behavioral arena, to which the mice were already habituated for a lengthy period (but not exactly the same for each mouse) before we recorded sufficient HD cells, it is not clear how much more in depth than previous studies we could investigate the learning reliability; (2) more detailed camera tracking of the mouse visual field would also be better suited to explore the relationship between reliable and unreliable cues; (3) denser RSC ensembles, similar to what the Sit and Goard 2023 study have reported, should be recorded.

2. Can you rule out potentially confounding effects of recordings obtained during early or late phases of the repeated exposures to the visual stimuli, that would affect the coherence between the AD and RSC HD representations? Are there more or less HD cells over time, when the animals have more prior experience with the environment?

We present in Author response image 5 an additional analysis by mouse where we track the number of recorded HD cells over time (A for ADn, B for RSC). Mice were also habituated to the environment for several days, even weeks before the recording sessions, while tetrodes were lowered until a substantial number of HD cells was detected. We note that some changes of the number of HD cells can also occur as a result of moving the tetrodes between sessions, and we refer the reviewers to Supplementary File 1 that tracks tetrode movements and several other parameters. Therefore, between the change in tetrode location and the already lengthy exposure to the arena, pinning the relationship between the number of HD cells to environmental exposure would be difficult in this dataset.

**Author response image 5. sa2fig5:** Number of HD cells over time. (**A**): Number of HD cells recorded in ADn over trials plotted by mouse. As in Rebuttal Figure 4, the open markers are trials rejected by the threshold for decoding accuracy in comparison with the decoded HD from shuffled spikes or by a decoded rotation less than 17.2°. Green indicates the large cue rotations, purple the small cue rotations. (**B**): Number of HD cells recorded in RSC over trials, as in **A**.

3. To help the reader to better understand how the experiments were structured, I suggest including an overview table, indicating, per mouse, for each of the 12 mice of this study:Mouse id;Tetrode positions in Suppl Figure 1;Which experimental protocols were run;How many sessions/trials;How many units in AD and/or in RSC were recorded;How many of them were HD or nonHD cells;Number of possible AD – AD pairs;Number of possible RSC – RSC pairs;Number of possible AD – RSC pairs.

We thank the reviewer for the suggestion, as it will clarify several details about the study and the dataset. We have included in Supplementary File 1 the metrics requested for each session and we will make this table available on Dryad together with the data for interested readers. Because we have not looked at the connectivity between pairs from the same region, we have not included those numbers – we do not believe it is within the scope of this manuscript or that would contribute to testing the hypothesis of the relationship between the two regions. Rather it might answer general local network arrangement, which, especially for a large region like RSC, might need higher density recordings.

4. Often times persistence of directional firing in the dark is used as a criterion for Head direction firing, and it may be useful to distinguish HD cells from visually responsive cells for example. I saw that 3 mice were tested in the dark (Figure 3), does it mean 9 mice were not? (How sure would you be to qualify directionally tuned cells as HD cells, and not visually responsive cells, if a dark condition was missing?)

We thank the reviewer for raising this point – and agree this may be a shortcoming of the study. Indeed, because not all mice were tested in dark condition, we cannot use this criterion for identifying HD cells and we rely only on directional tuning properties described. We have however restricted the criterion for directional tuning by making sure that directional information calculated as bits/spike, resultant and concentration parameters are above 98% of the shuffled control, thus raising the threshold from the previous assessment. Moreover, only cells for which these conditions were met in two distinct trials within the session were selected as HD cells. In Figure 1 – Supplement 3B and C we show the distribution of these parameters. As an additional quality assurance, we also assessed the stability of the preferred firing direction (Figure 1 – Supplement 3F) between the first and second half of a trials, as suggested elsewhere. We also showed that the position of the cue was not a determinant of the preferred firing directions of HD cells (Figure 1 – Supplement 3D). Additionally, we show in Author response image 6 the HD information calculated as bits/spike plotted against the egocentric angle with respect to the cue for our selected HD cells in ADn and RSC, to address the potential confound of selectivity to the visual cue. Given that the egocentric information falls below the unity line for the vast majority of ADn HD cells, we are confident in our HD classification. RSC HD units are more variable, as already indicated throughout the manuscript and as suggested in the previous literature, likely owing to the conjunctive nature of associative cortical neurons, and therefore a clear-cut distinction is not as apparent as in ADn.

**Author response image 6. sa2fig6:** Egocentric information of HD cells. Left, ADn (n=150) HD directional information (bits/spike) of classified HD cells vs egocentric directional information with respect to the angular position of the cue, with both axes on the logarithmic scale. Units for which egocentric information is not above 95% of the shuffled spikes were marked as open circles. Right, same analysis applied to RSC HD cells (n=73).

Furthermore, the criteria for HD cells typically include cue control. Because of the lack of a disorientation period during the cue rotation, this may be difficult to affirm for the protocols used here. Do you think it is still justified to qualify all the directionally tuned neurons as HD cells?

In the experiment design, we did not include a disorientation protocol for technical reasons but also to allow a training and testing of the HD decoding on stable cue conditions and assess directly the effect of the rotation on the HD representation by looking at the decoded error. We agree that not including this criterion might be a shortcoming for HD classification, especially of RSC units, which have been shown in previous studies to have different responses to cue and environmental manipulations (Chen, et al., 1994; Jacob, et al., 2017). However, we show correlated mean HD rotations calculated from the tuning curves in dual area recordings in Figure 1H (where trials from the same ensembles but different size of rotations were averaged), even though very few sessions were included, given that RSC HD cells are sparse. This suggests that on average HD cells from the two regions respond similarly with cue rotations and that our classification based on information, resultant and concentration parameters, and the peak stability quantifications largely captures HD cells.

Did you examine if units identified as HD might change and become nonHD (or vice versa) across different recording conditions?In the dark recordings, do you find that some directionally tuned cells become silent, and might those be cells responding to the visual cue?

We have compared the resultant of HD cells tuning curves during light condition with those from the first 2 minutes of dark conditions (before significant drifting could be observed) to address this point in Figure 1 – Supplement 3J, and show that this parameter does not significantly change as assessed by Wilcoxon Signed-Rank test. Similarly, we have compared the peak firing rate in Figure 1 – Supplement 3I and show that it is largely stable across conditions.

5. Please show more examples of tuning curves of HD and nonHD cells in addition to those in Figure 1B. I would find it more convincing and also a helpful resource, for reference, to see the range of different shapes of the tuning curves in AD and in RSC, their width, and peak firing rates.How many HD cells have more than 1 peak (suppl Figure 2A), in AD and in RSC?

We have added more examples of HD tuning curves both for RSC and for ADn in Figure 1 – Supplement 2 and have added the number of peaks for ADn and RSC HD cells in the main text.

6. All head direction cells were pooled together to produce suppl Figure 2D. In an ideal setting, one would expect the diagonal of preferred peak firing directions to be straight and to show uniform coverage of the 360{degree sign}, at least for ADn. This is not entirely the case. Could some of the HD cells be cells that are tuned to the visual cue? Can you indicate the cue position(s) on the graph, and might they be overrepresented?

We thank the reviewer for suggesting this analysis to strengthen the point of the independence of the HD cells from the visual tuning. We have added the cue position for the reference trials from which the tuning curve is calculated to the original Supplementary Figure 2D – which is now Figure 1 – Supplement 3D. We observe no overrepresentation of tuning.

7. Decoding: how many neurons were included? A range is given in the Discussion section, line 423, this information should rather be moved to the methods, and the actual number of simultaneously recorded units for each ensemble (Figures2, 3, S6) indicated in the results (or figure legends). How may this number influence the accuracy of the decoding?

Details about the number of HD cells as well as total number of neurons recorded can be found in Supplementary File 1. We also present in Author response image 7 the relationship between accuracy of decoding and number of HD cells recorded. As expected, the decoded error increases with fewer HD cells and also fewer total number of neurons, as captured by the exponential relationship.

**Author response image 7. sa2fig7:** Decoding accuracy over the number of cells. Left, decoding accuracy of ADn ensembles (n=36) over the number of ADn HD cells (light blue line exponential fit and 95% confidence interval, filled markers) and over the total number of recorded ADn cells (dark blue line exponential fit and 95% confidence interval, open markers). Right, same plot but for RSC (n=29), with bright red and dark red, respectively, to indicate the relationship over the number of classified HD cells (filled markers) and total number of units (open markers).

8. What is the firing frequency in light and in dark (Figure 3, suppl Figure 6), and is decoding still reliably carried out at low firing frequencies?

We have included in Figure 1 – Supplement 3H a comparison of all the mean firing rate of the recorded ensembles and in Figure 1 – Supplement 3I of the peak firing rate of unique HD units in both regions between cue on and cue off conditions. While the mean firing rate of the whole ensembles marginally change only for RSC ensembles, the peak firing rates of HD units are slightly decreased in darkness as opposed to cue on conditions in both regions (p=0.0027 for ADn, p=0.0004 for RSC, Wilcoxon Signed-Rank test). To quantify the decoding accuracy in darkness, in a similar approach as that taken in Figure 2 – Supplement 1G and H for the cue rotations, we compare in Author response image 8 the decoded drift calculated every 2 min during cue-on and cue-off against that of the cells’ tuning curves, calculated from the preferred peak during the first stable period. We observed that, while during cue-on both for ADn and RSC the values are clustered around 0 (such that circular cross correlations are close to 0 and with p values>0.05), during cue-off there is more drift outside of 0, and the decoded and the mean ensemble drifts from the tuning curves are significantly correlated (p<0.0001). The correlation is more evident for the ADn HD ensemble, as more values are concentrated along the unity line. The slight decrease in firing rates in darkness therefore does not affect the accuracy of the decoding.

**Author response image 8. sa2fig8:** Accuracy of decoded drift in darkness. Top, the decoded drift from individual ADn trials during cue-on (left) and cue-off (right) periods, calculated every 2 minutes, plotted against that calculated from the tuning curves of HD units. Bottom, same analysis but for RSC HD ensembles.

9. Figure 2C. It may be misleading to label the Y axis as a 'decoded error' with respect to a visual cue when the visual cue might not function as a visual landmark (see point 1, the mouse might rely more on self-information, proprioceptive and vestibular, than the visual cue). Figure 2C would benefit from showing more time before 0 (baseline).

We thank the reviewer for helping us improve on the readability of Figure 2 and have added more baseline to the examples in Figure 2C. We also want to clear any confusion within the labeling in this figure and have adjusted it to label in green the Cue rotation and time 0 rather than just “Cue”, and adjusted the text accordingly, to clarify that the decoded HD error is not calculated with respect to the cue position but rather as a subtraction to the tracked HD.

10. Line 359 states that AD-to-RSC connectivity was divergent, but in Figure 4H it appears that four differently tuned AD neurons, in blue, contact two (same?) RSC neurons (same-shaped tuning curves 1-2, and 3-4). Is this a mistake? Also, can you confirm that the RSC (red) traces all qualified as HD units (it looks quite untuned by the eye)?

We thank the reviewer for the correction and have removed the statement about “divergent connections”, as indeed a small number of ADn units make connections to the same RSC unit (as exemplified in the previous Figure 4H, which was not a mistake). We have revisited the dataset to provide more examples.

11. In addition to the connectivity rates indicated in Figure 4, I would suggest also including the numbers for ADn-ADn and RSC-RSC pairwise connectivity rates, for comparison, and for the sake of completeness.

As stated above, we did not run the within region functional connectivity pairs as it does not provide evidence of the relationship between ADn and RSC.

12. line 274, "highly synchronized" – please indicate the time scale.

We have added 20 ms time scale in the text and have removed the word “highly”.

13. It is not easy to read the color codes in some instances, including Figure 4G FS vs RS; especially when histograms overlap, also in Figure S2BC and Figure S6A.

We thank the reviewer for pointing this out and we have separated the overlapping histograms in the original S2BC to improve the visibility. We have changed the colors for Figure 4G FS vs RS and revisited the original Supplementary Figure 6A, now Figure 3F.

14. Figure 4 rabies tracing: There is only one example for each, please indicate n = 1, unless there are more experiments that are not shown (in which case, please show them in the supplementary data). In B, why are the presynaptically labelled cells in RSC red? Shouldn't they be green?

We have now added a second rabies tracing experiment for RSC in Figure 4 – Supplement 1A and a red retrobeads experiment in ADn in Figure 4 – Supplement 1B to respond to point 6 of the necessary revision points and reviewer’s 2 concern of viral tropism. To avoid any confusion we have noted in the figure legend that different constructs with opposing colors were used for the rabies tracing experiments, namely pAAV-syn-FLEX-splitTVA-EGFP-tTA + pAAV-TREtight-mTagBFP2-B19G and then (EnvA)SAD-ΔG-mCherry (Wickersham) for the ADn rabies tracing, thus giving rise to red pre-synaptically labeled cells, versus AAV1-hsyn-DIO-TVA66T-dTom-CVS-N2C(g) then EnvAdGCVS-N2CHistone-eGFP (Allen Institute) for the RSC rabies tracing, as we found out this latter construct was not working for ADn rabies tracing.

Figure 4G: Is there really similar tuning between connected Ad and RSC HD units (line 451)? Even if the circular mean of the RS peak differences is at -7.44{degree sign} (Figure 4G), very few pairs seem to have such a small difference in preferred head direction.

We have revisited the dataset and the HD selection criterion, resulting in refinement of our HD unit pairs and narrowing of the overall peak differences (mean peak difference = -5.4°). We have adjusted the text to more carefully argue that there is a bias, however with some variance.

15. Suppl F1b: Why are there so many tetrode locations in RSC, the text said 11 mice were recorded in RSC. Give AP levels for thalamic sites, to help the reader to situate the lesions.

We apologize for the confusion, and we have adjusted Figure 1 – Supplement 1B accordingly, with different colors for different mice. All RSC tetrode locations (obtained from the lesions) from all 11 mice with tetrodes in RSC are plotted in the figure. We have added the coordinates of each lesion in Figure 1 – Supplement 1A.

16. Mouse numbers:Line 74, 9 mice were simultaneously recorded ;Line 97, 8 mice… which is correct?

The text was changed accordingly. One mouse with simultaneous ADn-RSC recording did not pass the threshold for 8 RSC units to be included in the decoding dataset, but it does contribute to the spike correlation pair numbers and in the tetrode locations figure.

Reviewer #2 (Recommendations for the authors):1. The authors state that 'our results provide direct evidence against the hypothesis that visually guided updates in the HD reference would first appear in the RSC' (page 17 lines 416ff). This is indeed a strong statement, but I am not sure it is entirely supported by the authors' data. I wonder if a temporal offset in cue-rotation responses between AD and RSC HD cells might have gone undetected in the authors' study (see also point 2 below). For example, the visual update drive – in the form of e.g. an excitatory input from RCS that realigns HD cells in the AD- could be difficult to detect, considering the temporal resolution of the authors' experimental design. Hence, it is difficult to prove a 'negative result' here. The authors refer to two possible limitations (page 20 lines 491-492): 'cue devaluation with repeated trials' and 'temporal control of the cue rotation'. Is it conceivable that these limitations might have prevented the observation of a temporal delay between the shifts in HD representations between retrosplenial and AD?

We want to first clarify that the two limitations mentioned in the final section of the conclusion apply to the characterization of landmark encoding cells, whose contribution to internal HD representations is not addressed in this study, but would indeed be of interest in future recordings. Specifically, a tight control of the entering of the cue in the visual field of the mouse (for example in a head fixed setting on a rotating platform, such as that described in Sit, and Goard, 2023) would most likely be necessary in characterizing the firing fields of landmark-encoding cells. We have however eliminated this sentence to avoid confusions.

We do not believe those limitations would prevent the observation of a temporal delay between HD representations because: a) if there was no rotation of the HD reference then the trial was excluded; b) because of the slow time course of the rotations a temporal offset would most likely be observed regardless of the exact (+/- 500 ms) control of the visual field. As stated in the discussion, it is unclear what behavioral conditions lead to a slow realignment of the HD reference in contrast to the Zugaro et al. 2003 study, where rapid realignment (80 ms) was observed, but slower timescales have been reported by Knierim et al. 1998 and by Ajabi et al. 2021. How the time scale of the realignment is related to the activity of RSC (or postsubiculum, as an alternative brain region for visual/HD integration) is unknown, but the Ajabi et al. 2021 study related the time course needed to differences in ADn network gains, with an initial spike possibly reflecting visual inputs.

We agree that the 20ms temporal resolution prevents the observation of a temporal offset occurring at shorter timescales, for example that of direct excitatory drive between the two regions. To observe this case, enough deep layers ADn-projecting neurons have to be recorded and these neurons have to encode HD to allow accurate decoding specifically from this ensemble at 1 ms resolution. Alternatively, one would have to search for specific combinations of ADn-RSC HD cells pairs with the same tuning and cue rotations events in which the mouse keeps the head still in the preferred orientation (as described in Zugaro et al. 2003). It is unclear whether the neurons described in Jacob et al. 2017 (as indicated in point 2) would be directly involved in conveying updated HD information to ADn, given that most RSC presynaptic neurons to ADn are found in granular and not in dysgranular cortex. However, they could be involved in locally updating HD (Page et al. 2018), as suggested by the study of Sit and Goard 2023, where the realignment to repeated rotations was supported by a subpopulation of cells with distinct firing fields, likely through inputs from both external visual and HD inputs and local RSC inputs carrying specific spatial and visual information. A more detailed study of the circuit connectivity and the operations carried by individual RSC circuit components would answer these questions and resolve whether these visual/HD transformations can occur at a smaller temporal resolution than the one we used in this study.

We also want to point out that in addition to local and long-range inputs to RSC carrying information necessary for these transformation, previous literature has shown that HD reference update in behavioral settings similar to that described here depend on postsubicular activity (Goodridge et al. 1998) and on cortico-mamillary inputs (Yoder et al. 2015), whereby HD update can occur upstream of thalamus. This evidence points to a coordinated action of multiple brain regions to keep the HD in register across the brain, suggesting that substantial HD reference offsets are unlikely to be observed when decoding from the large ensemble levels.

Following the main suggestions and the comments of the other reviewers, we have (1) systematically checked our dataset and set stricter criteria for including trials and units; (2) we have averaged across trials with overlapping ensembles to perform statistical assessments on independent samples; and (3) we have offered an alternative method for checking the 0-ms time lag of HD coordination between the two regions in Figure 2 – Supplement 3. Our results are similar to those previously presented, suggesting the robustness of our data and methods. Moreover, we have decoded the internal HD representation of both regions with an alternative method, namely a LSTM network with only one hidden layer. Even in this case, the coordination between the decoded HD of the two regions by calculating the cross-correlation occurs at a 0-ms time lag. Because of the higher sensitivity of the difference between decoded errors to moment to moment variability in the 5s segments and the slightly lower decoder accuracy from the LSTM prediction, more variable distributions of time lags of minimum median error were observed; however, most of the traces had troughs at 0 ms time lags.

2. It is surprising that the authors recorded in the retrosplenial cortex, but do not comment upon nor describe bidirectional HD cells (Jakob et al., 2017). Since these neurons are regarded as prime candidates for the sensory-HD integration the authors aim to study, I think they should focus their analysis on these firing patterns specifically.

We have extended the discussion to consider the potential sensory-HD integration neurons proposed in the Jacob et al. 2017 study as candidates for landmark-based update of HD representation in RSC. However, we believe that these bi-or multidirectional firing patterns, as discussed in Jacob et al. 2017, are coupled to specific environmental configurations, i.e. the two distinct symmetrical rooms that can be closed and through which the animals can continuously navigate. Similarly, in the studies of Sit and Goard, 2023 and LaChance et al. 2022 bidirectionally tuned cells were identified in retrosplenial and postrhinal cortex, respectively, in environments with symmetric cues, and suggested this class of cells could discriminate between visual cues and process landmarks that anchor the HD. However, because these set-ups are not proposed in our study, a search for these patterns in our dataset does not seem feasible.

3. The tuning curves of HD cells in the AD nucleus are very broad (e.g. Figure S2E, Figure 1B, E), which seems to be at odds with prior literature, and this is potentially concerning. I wonder if this might be accounted for by the authors' definition of HD cells, and/or by spike sorting quality. The authors should try to quantify and exclude systematic spike sorting issues. As a 'control' they should maybe apply more stringent inclusion criteria, restrict the analysis to 'good' sharply tuned and nicely sorted HD units, and see if their conclusions hold for this refined dataset as well. They should also comment on whether and how (possibly suboptimal?) sorting might have impacted the 'negative result' of the present study, i.e. the fact that against predictions from prior literature, the authors did not find evidence for a visual update drive in the RSC cortex.

We provide in Figure 1 – Supplement 2A-C additional tuning curves of selected HD cells from different mice in ADn and RSC for two distinct angular positions of the visual cue, together with the corresponding spike autocorrelation histograms and the waveforms on all 4 channels of the tetrode where the units were detected. In Figure 1 – Supplement 2D we also show the distributions of the noise overlap and isolation metric, as directly extracted from spike sorting, of all our HD and non HD cells and we highlight: (1) the isolation metric is strongly skewed toward the maximum (on a scale from 0 to 1); (2) the noise overlap, in contrast, is skewed toward 0, with a small subset of units having a 0.0-0.4 overlap; and (3) these metrics are largely overlapping across all 4 categories.

We have refined the HD unit selection with stricter criteria: in addition to raising the threshold of resultant length and spike information to 98% of those obtained from spike shuffling, we included the concentration of the tuning curves as an additional parameter and, together with the sampling of unique ensembles following the tetrode movement more closely for both RSC and ADn, indeed the percentage of RSC HD units decreased and that of ADn HD units slightly increased.

4. The authors state that "the RSC is not wired to drive visual reference update" (page 17 line 413). However, connectivity RSC◊ AD has been demonstrated in several prior studies, and this is somewhat not recapitulated by the authors' tracing experiments with rabies viruses (I wonder if this finding needs to be confirmed with more conventional tracers, to exclude suboptimal tropism or synapse-specific effects of the viral transynaptic tracing, e.g. Rogers and Beier J Neurosc Methods 2021). The authors' statement is also supported by functional connectivity data (cross-correlations). The 'apparent sparsity' of connectivity could however be biased by cellular sampling since most thalamic projecting neurons are expected to be found in the deep layers (mostly L6) which seem not to have been systematically sampled with tetrode recordings.

We have expanded our discussion to clarify this point, and rephrased the quoted sentence to express that RSC, while playing a major role in encoding landmarks and translating between internal and external spaces, in our experimental paradigm and through our data analysis does not seem to anticipate, at the ensemble level and at the time resolution of 20 ms, the HD reference with respect to ADn during cue rotations.

We have provided an alternative anatomical tracing strategy to overcome potential poor tropism of the virus, or, as discussed in point 6 of the essential revisions, potentially low number of infected starter ADn neurons. With retrobeads injected into ADn, we show results in line with those from previous literature (Shibata, 1998), with deep layers, especially in granular RSC, labeled. As for the sparsity from the functional connectivity in the RSC-to-ADn direction, we agree with the reviewers that a more focused sampling of layer 6 of RSC and possibly also in more posterior RSC closer to the postsubiculum would have revealed more connections. However, based on the ensembles we have recorded, with 9 tetrodes out of 48 in the mice with simultaneous ADn-RSC recordings, the internal HD we have decoded from the two regions and the temporal coordination are supported by the functional connections we have observed. Furthermore, these connections were determined with a high threshold (99.9% probability from a convolved distribution, used as baseline).

5. The authors state that they have performed 'widespread sampling of RCS locations' (page 18 line 449). This is to some extent true; however, the retrosplenial cortex goes well beyond the recording locations sampled by the authors. I think the authors should reword these statements and acknowledge the possibility that other subfields of the retrosplenial cortex (which were not extensively sampled by the authors) could in principle be responsible for the visual update drive to the thalamus.

We agree with the reviewer that particularly more posterior locations in RSC, where the granular portion borders with postsubiculum, have not been substantially sampled. We have reworded our discussion to highlight the need of extensive recording of RSC in the anterior-posterior axis in future studies to test their roles in these visual realignment tasks, especially given that RSC is already known to display a gradient in encoding egocentric versus allocentric navigational variables (Hennestad et al. 2021). As mentioned in the discussion, there are however several different pathways through which HD in ADn can be updated: 1) direct postsubiculum to ADn connections, and 2) cortical (postsubiculum) to LMN connections. It is unclear whether different behavioral conditions would prioritize one pathway as supposed to another or whether, regardless of the behavioral demands, a coordinated HD representation is ensured at different levels in the HD circuitry through multiple connections.

Reviewer #3 (Recommendations for the authors):Decoding errors: Average decoding error in RSC seems very high – 90.83 degrees for fast head turns seems like a chance level and I am not sure if it is still appropriate to talk about successful decoding of HD in that case. Since the quality of decoding is critically dependent on the number and tuning of HD cells recorded in each session, could the authors provide evidence that in each session included in the dataset both ADN and RSC decoders perform significantly better than chance as estimated from spike shuffles?

We want to clarify that in the original version of the paper (Supplementary Figure 3C) we presented the degree value corresponding to the 75 percentile of the error distribution. We realized that the median of the absolute decoded error is a better metric for this analysis and have thus adjusted the graphs to reflect this (now Figure 2 – Supplement 1D), which has brought the accuracy metric to lower values. We also followed the reviewer’s suggestion to eliminate sessions for which the median decoded error was higher than the 10th of 100 values calculated from shuffled spikes (shown in Figure 2 – Supplement 1B).

Examples in Figure 2A show many gaps in the tracked HD of the mouse, which to me indicates the sub-optimal quality of the behavioural tracking. This is especially important for analyses of decoding errors like the one in Figure 2D that shows that internal HD representations in ADN and RSC are coordinated at zero lag (+/- 20ms). The observed zero-lag peak could be instead explained by errors in behavioural tracking dominating the analysis, which would affect both representations simultaneously and show spurious zero-lag positive correlations. However, coordination could also be shown by computing the difference between internal HD decoded from ADN and RSC at different time lags, without reference to the HD tracked behaviourally. Could the authors include this sanity check in the manuscript?

There might be at times gaps in the HD tracking if the sensors on the headstage are obscured, for example during grooming bouts, or by the connecting cable. HD is tracked through the HTC Vive Base Station system detection from 3 receivers in 9 mice – 2 in 3 mice – placed on top of the headstage, at a 30 Hz frequency. To reach the 50Hz sampling resolution we use throughout the manuscript, we interpolate between data points. We have provided a new example in Figure 2A and have not median-smoothed the HD tracking as we did in the previous version of the figure.

As for the zero-lag between HD representations, we followed the reviewer’s suggestion, and plotted the difference between the decoded HD of each region in Figure 2 – Supplement 3B. Specifically we have taken the absolute median of the difference between 5 s segments and temporally offset them up to +/- 5 s, and repeated this to cover the length of 75s of decoded trace. We note that there are however some factors that contribute to the high variability and would potentially obscure effects in the scenarios of either region leading the HD reference change: 1) the direction of the cue rotation; b) the direction of the head rotations (clockwise vs counterclockwise). For these reasons, initially we took the cross correlation of the decoded errors. The sign and the magnitude of the angular velocity are the major factor leading to large error differences in the -180 to 180 range at around 5s temporal lags, therefore we decided to take the absolute median error. Finally, to combine our results in the face of the different directions of the rotations, we made all the decoded errors have the same sign. We can see from the histogram of the troughs from unique ensembles that the majority lies at 0, recapitulating the temporal cross correlation results.

The percentage of HD cells in RSC reported in the literature is generally low (~10%), but those RSC HD cells often show very narrow HD tuning. Yet, judging by the examples, the HD tuning of RSC cells recorded in the study seems worse than expected. Could the authors provide some statistical measure of HD tuning stability for each cell (i.e. correlating the first and second half of the baseline recording) as a sanity check? Canonical HD cells should show very good tuning stability under constant conditions.

We have included the suggested analysis of peak stability in Figure 1 – Supplement 3F and indeed found that the peak is quite stable (circular correlation = 0.93 for ADn, n=150, for RSC 0.79, n=73, p<0.0001 for both). We have also revised the HD selection criterion to be more stringent. As suggested by other reviewers and by the essential revision, we have also added more examples in Figure 1 – Supplement 2.

Supplementary Figure 2F seems to show an overrepresentation of HD cells with similar preferred directions, especially so in RSC. If this is indeed the case, I am wondering if this apparent HD tuning could be explained by any other behavioural variable that in this session happens to be correlated with HD, as in my experience sometimes happens when a tuning curve shows weak HD tuning. I find that such spurious HD tuning often disappears if only epochs when the animal is moving are included in computing the HD curve, could the authors demonstrate that HD tuning does not change if stationary epochs are excluded?

We have rerun the HD cells selection by refining the criteria (namely selecting bits/spike, resultant and concentration parameters above 98% of those from shuffled spikes for both ADn and RSC units in 2 distinct segments of the recording sessions) and by selecting only time points within a period of stable cue where the animal is moving, as here suggested.

Could the authors provide the breakdown of overall cell numbers recorded per animal (including nonHD/HD cells), for example as a table?

We have provided in the essential revision point 4 the Supplementary File 1 summarizing total number of cells and number of HD cells in each region per recording session, per mouse, as well as the number of ADn-RSC pairs tested for functional connectivity and the tetrode movements.

Are there any differences in HD modulation across layers and sub-regions of RSC?

We agree with the reviewer that this would be an interesting avenue of research to better understand the organization of RSC. However, we think that this characterization of HD within cortical layers would be better addressed with high density laminar probes where layer identification is more accurate and sampling much higher.

Figure 1J shows that HD cell receptive fields exhibit both clockwise and counterclockwise rotations, do these correspond to the clockwise or anticlockwise rotations of the cue, or is the realignment not dependent on the direction of cue rotation?

No, indeed sometimes the ensemble/decoded rotation is in the opposite direction or overshoots (as can be seen in Figure 2 – Supplement 2B) and, as a consequence of the circularity, it seems the HD representation goes in the opposite direction as the imposed cue rotation.

The manuscript often uses a number of trials as their sample size for statistical analyses and the methods state that tetrodes were regularly advanced, but there is no indication of whether multiple trials at the same tetrode position were included in the same statistical comparison (except for recordings '4 days apart' for the HD tuning and synaptic connectivity analyses). Multiple trials with a high likelihood of recording the same cell population should not be counted as separate samples when calculating statistical significance. The authors should clarify whether tetrodes were moved between recording sessions and, if that was not the case, correct their statistical analyses to, e.g. perform statistical tests on average values per recorded population over multiple sessions.

Indeed, unlike in the HD quantification and spike correlation analyses where we originally took recordings 4 days apart, all trials were included in the previous versions of Figure 2 and 3, whether they were within the same recording session (as shown in the examples) or in next day session. In our revised manuscript we have followed the reviewer’s suggestion and have adjusted Figures 2 DandE, 3D-G and Figure 2 – Supplements 1 and 3 and Figure 3 – Supplement 1 to average across sessions containing largely overlapping ensembles and perform statistical tests on unique ensembles following more closely the tetrode movements and changes in recorded unit numbers. Similarly, we have sampled sessions with different ensembles (or combinations of RSC-ADn ensembles for the functional connectivity) to present and perform statistical analysis for the data in Figures 1 and 4 and Figure 1 – Supplement 3 and Figure 4 – Supplement 2. We have adjusted in the figures and figure labels the identification of n to ensembles.

[Editors’ note: what follows is the authors’ response to the second round of review.]

The manuscript has been improved but reviewer #2 has some remaining issues that need to be addressed (mostly clarification of result interpretation and discussion), as outlined below.Reviewer #2 (Recommendations for the authors):In response to my concern #1, the authors state: "We agree that the 20ms temporal resolution prevents the observation of a temporal offset occurring at shorter timescales, for example, that of direct excitatory drive between the two regions". This potential limitation should be specifically acknowledged in the discussion, e.g. by stating that the temporal resolution of the decoding approach (20 ms) might have prevented the resolution of faster (monosynaptic) dynamics, which are approx one order of magnitude faster. Again, I believe that saying "our results provide direct evidence against the hypothesis that visually guided updates in the HD reference would first appear in the RSC" (lines 343ff) is a very strong statement. The authors provide evidence consistent with this hypothesis but do not formally rule out alternative updating mechanisms that might occur on faster timescales, i.e. beyond the authors' temporal resolution. Faster dynamics (<20ms) might have gone undetected in the authors' work. This limitation should at least be acknowledged in the discussion.

We thank the reviewer for raising this point to more precisely and carefully communicate our results in light of potential limitations. We have eliminated the sentence “Together, our results provide direct evidence against the hypothesis that the updated HD reference from visual cue rotations would first appear in RSC, at the ensemble level, and then be directly conveyed to ADn” in the discussion and, as the reviewer requested, added the sentence “However, the temporal resolution of the decoding occludes potential faster dynamics, at the scale of monosynaptic connections (ms range)” to acknowledge the limitation of our 20ms window.

The sentence in the abstract "with surprisingly little reciprocal drive in the corticothalamic direction" is not supported by the authors' data, nor by previous anatomical work showing strong RS deep layer-to-ADn connectivity. This is also clear from the authors' tracing data (from retrograde tracings, RS L6 shows up prominently). At some points, the authors make granular vs agranular RS distinctions (e.g. lines 377ff), but this is not done consistently throughout the manuscript, and the above sentence in the abstract generally refers to RS. The authors also refer to "asymmetry" in RS – ADn connectivity; I do not see what the authors mean by 'asymmetry', since the connectivity follows the classical thalamocortical rules (e.g. L6 back to thalamus). As for the functional data, more extensive sampling of RS L6 is needed to support claims about asymmetric connectivity. Hence, such strong claims (as the one in the abstract) should at least be moderated throughout the manuscript.

We have edited the abstract to tone down the concept of asymmetry in light of the L6 labeling in the retrobeads tracing, and have also changed the heading of the Results section accordingly. And in the discussion, we have specifically added the sentence “Further denser sampling of layer 6 in RSC, and more specifically in the granular portion, where ADn’s presynaptic partners in RSC are mostly found ((Shibata, 1998), Figure 4B, Figure 4 – Supplement 1B), are needed to clarify the role of the anatomically-identified corticothalamic projections in the visual updating of ADn-RSC HD reference”.